

# A new kentriodontid (Cetacea: Odontoceti) from the early to middle Miocene of the western North Pacific and a revision of kentriodontid phylogeny

Zixuan Guo[1] and Naoki Kohno[1,2]

[1] Graduate School of Life and Environmental Sciences, University of Tsukuba, Tsukuba, Japan
[2] Department of Geology and Paleontology, National Museum of Nature and Science, Tsukuba, Japan

## ABSTRACT

A new species of an extinct dolphin belonging to the kentriodontids, i.e., *Kentriodon sugawarai* sp. nov., is described from the upper lower to lowest middle Miocene Kadonosawa Formation in Ninohe City, Iwate Prefecture, northern Japan. The holotype of *Kentriodon sugawarai* sp. nov., consists of a partial skull with ear bones, mandibular fragments, and some postcranial bones. This new species shares five unique characters with other species of *Kentriodon*. In addition, the new species differs from other species of the genus in displaying a narrow width of the squamosal lateral to the exoccipital in posterior view, the dorsolateral edge of the opening of the ventral infraorbital foramen that is formed by the maxilla and the lacrimal or the jugal, and at least three anterior dorsal infraorbital foramina. Our phylogenetic analysis based on 393 characters for 103 Odontoceti taxa yielded a consensus tree showing all previously identified kentriodontids as a monophyletic group that comprises the sister group of the crown Dephinoidea, which in turn include Delphinidae, Phocoenidae and Monodontidae. Our analysis also indicates that the distinct innovation of the acoustic apparatus (i.e., 13 out of 29 derived characters are from tympanoperiotic) would have occurred in the ancestral lineage of the Delphinoidea (sensu lato) including the monophyletic Kentriodontidae during their initial evolution and diversification.

## INTRODUCTION

Dephinoidea (i.e., Delphinidae, Monodontidae and Phocoenidae) has been thought to emerge in the early Miocene (*Gatesy et al., 2013*) and they are the most species-rich clade of living marine mammals in the world. However, their evolutionary origins are still puzzling. Proceeding in the dawn of Dephinoidea, small coastal odontocetes known as kentriodontids (*Barnes & Mitchell, 1984*; *Barnes, 1985*; *Muizon, 1988a*) attained a high diversity during the period between the early and the late Miocene (*Ichishima et al., 1995*; *Kazár & Hampe, 2014*; *Peredo, Uhen & Nelson, 2018*). This group had been considered to be placed among the stem delphinoids based on their primitive cranial morphologies and retention of several ancestral characters of odontocetes (*Barnes, 1978*). For instance, asymmetric nasals and

Corresponding author
Zixuan Guo,
guo_z@geol.tsukuba.ac.jp

premaxillae have commonly been observed in modern odontocetes. However, in part of the taxa referred to as members of kentriodontids, these bones are seemingly symmetrical. The interpretation of the evolutionary patterns of the Delphinoidea greatly relies upon the processes of morphological transformation in their stem group, while the phylogenetic relationships of such a stem group, presumably the kentriodontids, have remained debated.

In the initial stage of the studies on kentriodontids, they were considered as comprising a monophyletic family, i.e., Kentriodontidae (e.g., *Barnes, 1978*; *Barnes, 1985*; *Muizon, 1988a*; *Muizon, 1988b*; *Ichishima et al., 1995*). However, recent studies have advocated that 'kentriodontids' are paraphyletic and should be subdivided into several clades by a different combination of taxa within Delphinida (including Lipotidae, Inioidea, and Delphinoidea) (e.g., *Lambert et al., 2017*; *Peredo, Uhen & Nelson, 2018*; *Kimura & Hasegawa, 2019*). Because several additional species of 'kentriodontids' have been recently reported, and molecular phylogenies of the cetaceans have been established in the last decade (e.g., *McGowen, Spaulding & Gatesy, 2009*; *McGowen et al., 2011*; *McGowen et al., 2020*; *Geisler et al., 2011*), a more comprehensive reappraisal of the phylogeny of this group is necessary. In particular, *Peredo, Uhen & Nelson (2018)* redefined the family Kentriodontidae, only including *Wimahl*, *Kampholophos* and *Kentriodon*. However, some other phylogenetic studies (e.g., *Murakami et al., 2014*; *Tanaka & Fordyce, 2014*; *Tanaka & Fordyce, 2016*; *Tanaka et al., 2017*; *Lambert et al., 2018a*; *Lambert et al., 2018b*), using different character sets and data matrices displayed some kentriodontid taxa in a different phylogenetic topology. In other words, the relationships of taxa originally referred to the family 'Kentriodontidae' are still debated.

Here, we describe a new species of kentriodontid from the early to middle Miocene of Japan (Fig. 1). The holotype specimen includes a partial skull with well-preserved tympanoperiotics. We also reassess the phylogenetic relationships of kentriodontids and discuss on the evolution of Delphinoidea, including kentriodontids.

## MATERIALS & METHODS

### Nomenclatural acts

The electronic version of this article in Portable Document Format (PDF) will represent a published work according to the International Commission on Zoological Nomenclature (ICZN), and hence the new names contained in the electronic version are effectively published under that Code from the electronic edition alone. This published work and the nomenclatural acts it contains have been registered in ZooBank, the online registration system for the ICZN. The ZooBank LSIDs (Life Science Identifiers) can be resolved and the associated information viewed through any standard web browser by appending the LSID to the prefix http://zoobank.org/. The LSID for this publication is: urn:lsid:zoobank.org:pub:B0E9467F-CDD3-4AF4-83FE-40CE09D15700. The online version of this work is archived and available from the following digital repositories: PeerJ, PubMed Central and CLOCKSS.

### Anatomical terminology

We follow *Mead & Fordyce (2009)* and *Ichishima (2016)* for the terminology of the skull.

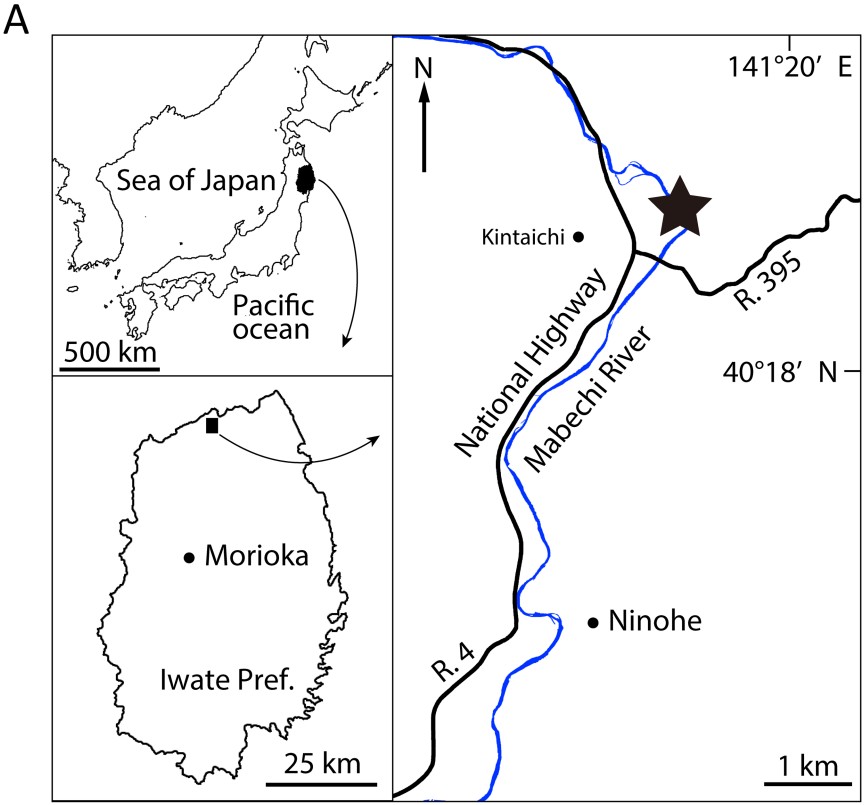

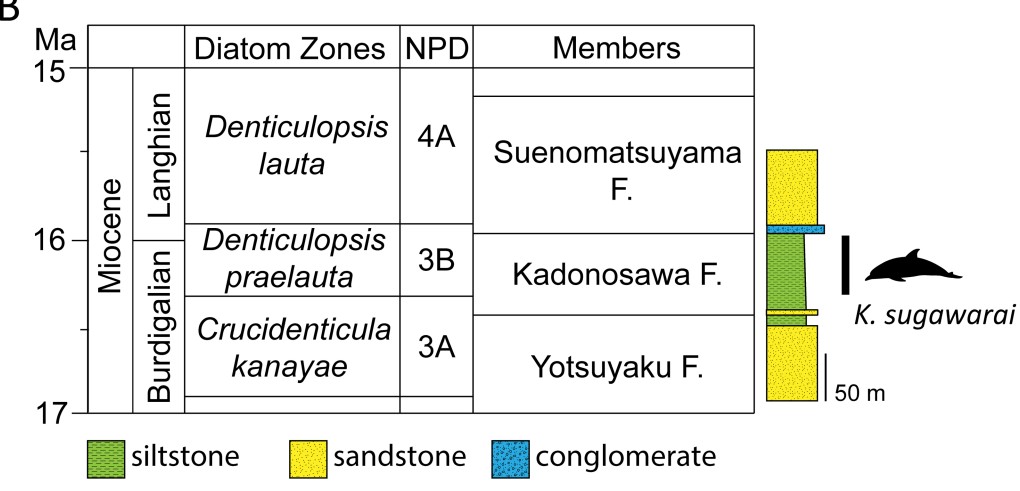

| Ma | | Diatom Zones | NPD | Members |
|---|---|---|---|---|
| 15 | Miocene | | | |
| | Langhian | *Denticulopsis lauta* | 4A | Suenomatsuyama F. |
| 16 | Burdigalian | *Denticulopsis praelauta* | 3B | Kadonosawa F. |
| | | *Crucidenticula kanayae* | 3A | Yotsuyaku F. |
| 17 | | | | |

siltstone    sandstone    conglomerate

*K. sugawarai*

50 m

**Figure 1** **Geographic and geological context of *Kentriodon sugawarai* locality.** (A) The type locality of *Kentriodon sugawarai* sp. nov., holotype, NMHF 999. (B) Left, diatom zone and stratigraphic diagram, modified from *TuZino & Yanagisawa, (2017)*. Right, stratigraphic column of the Mabechi River, Ninohe City, Iwate Prefecture, Japan.

## Phylogenetic methods

The phylogenetic position of the new species described here was analyzed based on a character list and the character matrix that stems from the works by *Tanaka et al. (2017)* and *Lambert et al. (2017)*. The character list and data matrix by *Tanaka et al. (2017)* derive from those by *Geisler et al. (2011)* via the addition of characters by *Murakami et al. (2012)* to understand interspecific relationships of the Phocoenidae within the crown Delphinoidea, and the subsequent modifications by *Tanaka & Fordyce (2014)*. The data matrix by *Tanaka et al. (2017)* included 87 taxa and 284 characters, but this matrix only included three kentriodontid taxa: i.e., *Atocetus iquensis Muizon, 1988b*, *Hadrodelphis calvertense Dawson, 1996a*, and *Kentriodon pernix Kellogg, 1927*. By contrast, the data matrix by *Lambert et al. (2017)* was also based on *Geisler et al. (2011)* and *Geisler, Godfrey & Lambert (2012)*, but it included 112 taxa and 324 characters, with 12 kentriodontid taxa: i.e., *Atocetus iquensis*, *Atocetus nasalis Muizon, 1988b*, *Delphinodon dividum True, 1912*, *Hadrodelphis calvertense*, *Kampholophos serrulus Rensberger, 1969*, *Kentriodon pernix*, *Lophocetus calvertensis Cope, 1867*, *Lophocetus repenningi Barnes, 1978*, *Macrokentriodon morani Dawson, 1996b*, *Pithanodelphis cornutus Abel, 1905*, *Rudicetus squalodontoides Bianucci, 2001*, and *Tagicetus joneti Lambert, Estevens & Smith, 2005*. However, the character set used by *Lambert et al. (2017)* for their phylogenetic analysis was originally elaborated for taxa within earlier branching clades of the Odontoceti (e.g., *Geisler et al., 2011*; *Lambert et al., 2017*; *Peredo, Uhen & Nelson, 2018*; *Kimura & Hasegawa, 2019*). Consequently, the focus of these two streams of studies on the odontocete phylogeny (e.g., *Tanaka et al., 2017* and e.g., *Lambert et al., 2017*) have not fully overlapped with each other, in other words, the included taxa of kentriodontids and character combination to analyze their phylogenetic relationships were far too different to each other. To solve these issues, we elaborated a phylogenetic dataset based on the combined characters and kentriodontid taxa from previous studies such as *Geisler, Godfrey & Lambert (2012)*, *Murakami et al. (2012)*, *Tanaka & Fordyce (2014)*, *Tanaka et al. (2017)*, *Lambert et al. (2017)*, *Peredo, Uhen & Nelson (2018)*, and *Kimura & Hasegawa (2019)*. The resulting data matrix that is used herein is based on 103 taxa, including almost all kentriodontids, and 393 morphological characters (see Supplemental Information), with a tree constraint based on the molecular phylogenetic studies of the extant cetaceans by *McGowen, Spaulding & Gatesy (2009)*, *McGowen et al. (2011)* and *McGowen et al. (2020)*. Regarding the kentriodontids, we included the following 15 taxa into our data matrix (see also Supplemental Information): *Atocetus nasalis* from the late Miocene of California, USA (*Muizon, 1988b*), *Delphinodon dividum* from the early Miocene of Meryland, USA (*True, 1912*), *Kampholophos serrulus* from the early Miocene of California, USA (*Rensberger, 1969*), *Kentriodon diusinus Salinas-Márquez et al., 2014* from the middle Miocene of Baja California, Mexico (*Salinas-Márquez et al., 2014*), *Kentriodon nakajimai Kimura & Hasegawa, 2019* from the middle to late Miocene of Japan (*Kimura & Hasegawa, 2019*), *Kentriodon obscurus Kellogg, 1931* from the middle Miocene of California, USA (*Kellogg, 1931*; *Barnes & Mitchell, 1984*), *Kentriodon schneideri Whitmore & Kaltenbach, 2008* from the middle Miocene of North Carolina, USA (*Whitmore & Kaltenbach, 2008*), *Liolithax pappus* from the middle Miocene of Maryland, USA (*Barnes, 1978*), *Lophocetus calvertensis* from the late Miocene of Maryland, USA (*Cope, 1867*), *Lophocetus repenningi*

from the middle Miocene of California, USA (*Barnes, 1978*), *Macrokentriodon morani* from the middle Miocene of Maryland, USA (*Dawson, 1996b*), *Pithanodelphis cornutus* from the late Miocene of Belgium (*Abel, 1905*), *Rudicetus squalodontoides* from the early to late Miocene of Italy (*Bianucci, 2001*), *Tagicetus joneti* from the middle Miocene of Portugal (*Lambert, Estevens & Smith, 2005*), and *Wimahl chinookensis Peredo, Uhen & Nelson, 2018* from the early Miocene of Washington, USA (*Peredo, Uhen & Nelson, 2018*).

The phylogenetic analysis was performed with TNT 1.5 (*Goloboff, Farris & Nixon, 2008*; *Goloboff & Catalano, 2016*). All characters were treated as unweighted and unordered, using the "New Technology Search" task to find minimum length trees 1,000 times, under a tree constraint based on molecular evidence from the extant taxa (*McGowen, Spaulding & Gatesy, 2009*; *McGowen et al., 2011*; *McGowen et al., 2020*).

## SYSTEMATIC PALAEONTOLOGY

CETACEA Brisson, 1762
ODONTOCETI Flower, 1867
DELPHINIDA *Muizon, 1988a*
KENTRIODONTIDAE Slijper, 1936

**Emended Diagnosis of Kentriodontidae**: Differing from other delphinidan families and superfamilies (i.e., Delphinidae, Phocoenidae, Monodontidae, Lipotidae, and Inioidea) in displaying the following suite of derived character states: premaxillae are compressed mediolaterally at anterior part of the rostrum (Chr. 3), the mesorostral groove is constricted posteriorly, anterior to the nares and behind the level of the antorbital notch, then rapidly diverging anteriorly (Chr. 7), anterior edge of the supraorbital process is oriented anterolaterally, forming an angle between 35° and 60° (Chr. 50), the dorsolateral edge of internal opening of the infraorbital foramen is formed by the lacrimal or the jugal (Chr. 58), the infratemporal crest forms a well-defined curved ridge on the posterior edge of the sulcus for the optic nerve (Chr. 64), the premaxillary foramen is located medially (Chr. 72), the alisphenoid is broadly exposed laterally in the temporal fossa (Chr. 160), suture between both the palatines and the maxillae is straight transversely or bowed anteriorly (Chr. 179), the external auditory meatus is wide (Chr. 225), the basioccipital crests form an angle of approximately 15–40° in ventral view (Chr. 229), the hypoglossal foramen is separated from the jugular foramen or the jugular notch by thick bone (Chr. 231), most convex part of the pars cochlearis is on the ventrolateral surface (Chr. 283).

*Kentriodon Kellogg, 1927*

**Emended Diagnosis of Genus**: *Kentriodon* differs from other genera of kentriodontids by the following unique combination of characters: the cheek tooth entocingulum is present

(Chr. 28); the dorsal edge of the orbit is low, either in line with the edge of the rostrum or slightly above it (Chr. 47); the position of the inflection of premaxilla is located in line with the posterior half of the supraorbital process or in line with the postorbital process of frontal (Chr. 109); in lateral view, the dorsal edge of the zygomatic process preserves a distinct dorsal flange or process near the anterior end, articulates with the frontal (Chr. 164); and the postzygapophysis is appearing as a crest, elongated dorsolaterally from anterior view (Chr. 328). In this regard, *Rudicetus squalodontoides* could also be included in this genus.

*Kentriodon sugawarai* sp. nov.
([Figs. 2–11](), [Table 1]())

**LSID**: urn:lsid:zoobank.org:act:0D209916-B472-44A7-B7AB-29682FA945C4
**Holotype**: NMHF 999, incomplete skull including most of the neurocranium and a proximal portion of the rostrum, one tooth, the right tympanoperiotic and malleus, fragments of the left and right mandibles, and the partial atlas.
**Diagnosis of Species**: *Kentriodon sugawarai* sp. nov. differs from *K. schneideri* by the convex occipital shield (Chr. 176). It differs from *K. pernix*, *K. nakajimai* and *K. obscurus* by the following characters: the dorsolateral edge of the opening of the ventral infraorbital foramen is formed by the maxilla and the lacrimal or the jugal (Chr. 58), at least three anterior dorsal infraorbital foramina (Chr. 65), anterolateral corner of the nasal lacking a distinct inflated process (Chr. 136), narrow width of the squamosal lateral to the exoccipital (Chr. 170), anterior level of the pterygoid sinus fossa is interrupted posterior to (or at the level of) the antorbital notch (Chr. 193), and the ventral edge of the anterior process of the periotic is clearly concave in lateral view (Chr. 245). Further differs from *Kentriodon hoepfneri* and *K. nakajimai* by the apex of the postorbital process of the frontal that is directed ventrally rather than posterolaterally, by the angle between anterior process of periotic and anterior edge of pars cochlearis that is nearly 90°. It differs from *K. nakajimai*, *K. diusinus* and *K. schneideri* by displaying a deep emargination of the posterior edge of the zygomatic process by the neck muscle fossa. It differs from *Kentriodon hobetsu*, *K. schneideri* and *K. pernix* by the transversely narrower exoccipital and by the maxillae that makes a deep fossae on each side at the vertex. It differs from *K. obscurus* and *K. pernix* by the aperture for the cochlear aqueduct that is transversely smaller than the aperture for the vestibular aqueduct. It further differs from *K. pernix* by having a shallower lateral furrow of the tympanic bulla.
**Etymology**: The species is named in honor of Mr. Kohei Sugawara, the former director of the Ninohe Museum of History and Folklore, for his longstanding contributions to geology and paleontology as well as local history of the Ninohe district, and as a sign of gratitude for his encouragement and assistance to both of us throughout this study.
**Type Locality**: The holotype was collected in the 1940s from a locality close to the Mabechi River, Ninohe City, Iwate Prefecture, Japan. Approximate geographical coordinates: 40°31′N, 141°31′E ([Fig. 1]()).

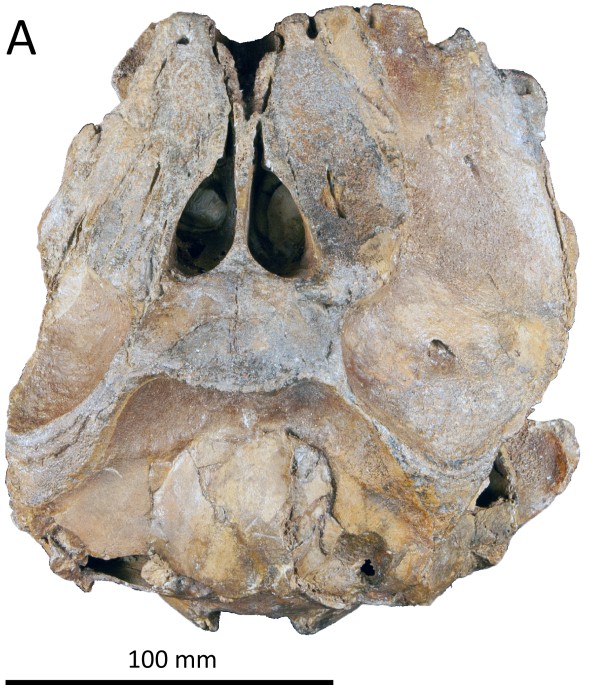

A

100 mm

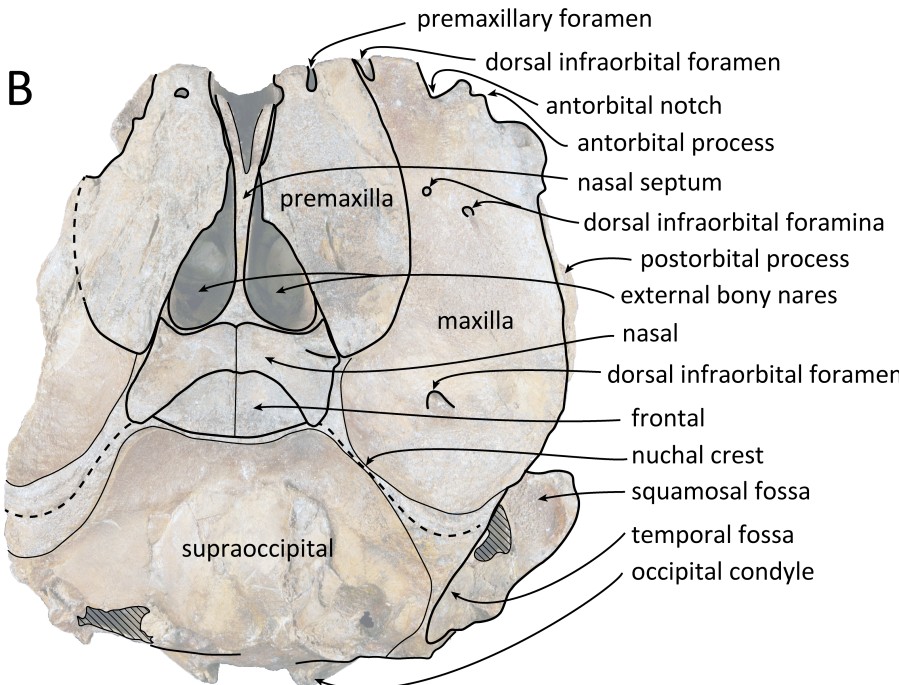

B

premaxillary foramen

dorsal infraorbital foramen

antorbital notch

antorbital process

nasal septum

premaxilla

dorsal infraorbital foramina

postorbital process

external bony nares

maxilla

nasal

dorsal infraorbital foramen

frontal

nuchal crest

squamosal fossa

supraoccipital

temporal fossa

occipital condyle

**Figure 2   Dorsal views of the skull of *Kentriodon sugawarai* sp. nov., holotype, NMHF 999.** (A) Photo. (B) Corresponding line drawing with anatomical interpretations. Scale bar equals 100 mm.

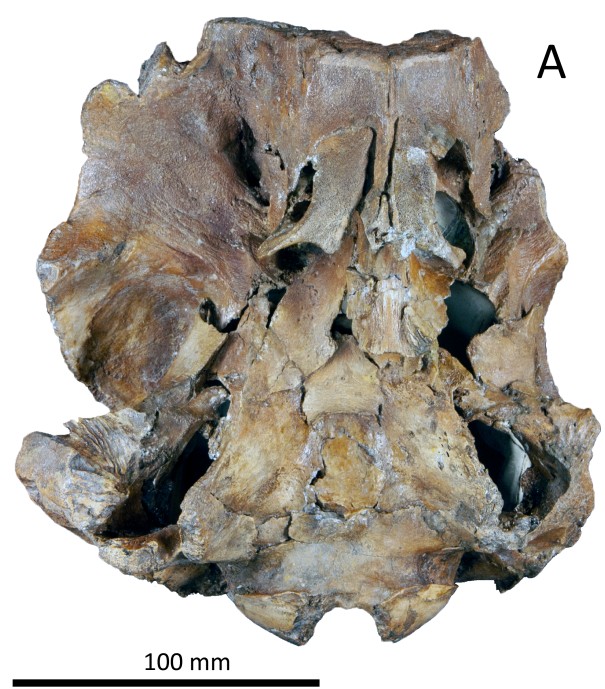

A

100 mm

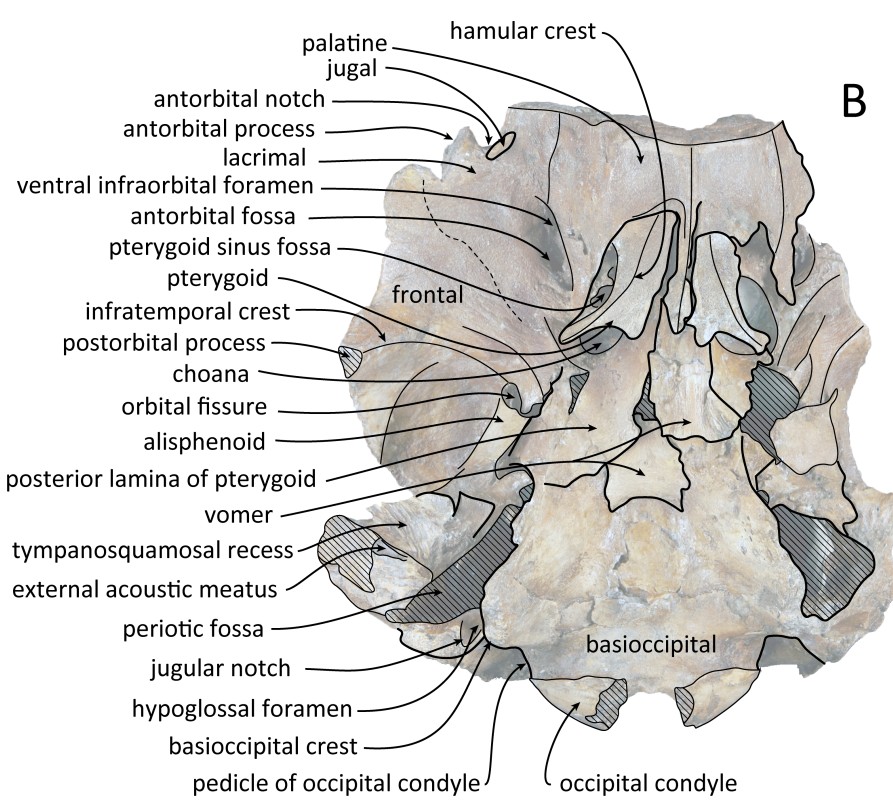

B

palatine
jugal
hamular crest
antorbital notch
antorbital process
lacrimal
ventral infraorbital foramen
antorbital fossa
pterygoid sinus fossa
pterygoid
infratemporal crest
postorbital process
choana
orbital fissure
alisphenoid
posterior lamina of pterygoid
vomer
tympanosquamosal recess
external acoustic meatus
periotic fossa
jugular notch
hypoglossal foramen
basioccipital crest
pedicle of occipital condyle
frontal
basioccipital
occipital condyle

**Figure 3** **Ventral views of the skull of *Kentriodon sugawarai* sp. nov., holotype, NMHF 999.** (A) Photo. (B) Corresponding line drawing with anatomical interpretations. Scale bar equals 100 mm.

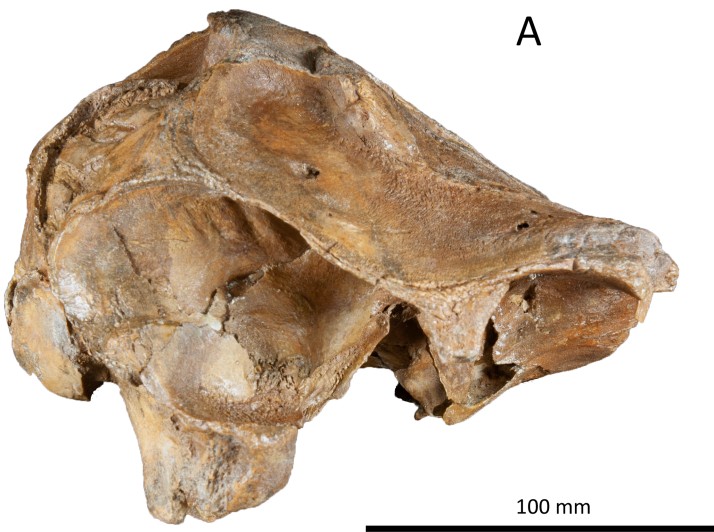

A

100 mm

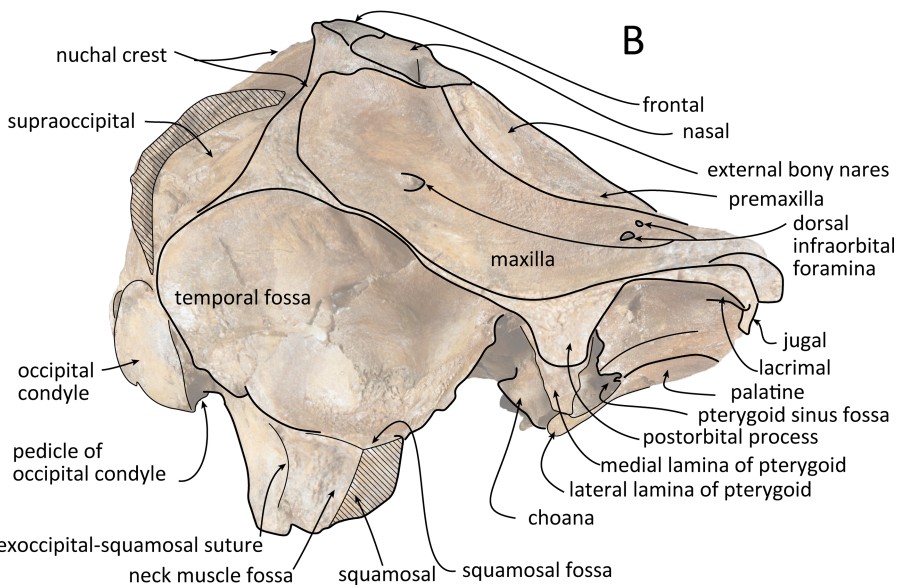

B

nuchal crest
supraoccipital
frontal
nasal
external bony nares
premaxilla
dorsal infraorbital foramina
maxilla
temporal fossa
jugal
lacrimal
palatine
pterygoid sinus fossa
postorbital process
medial lamina of pterygoid
lateral lamina of pterygoid
choana
occipital condyle
pedicle of occipital condyle
exoccipital-squamosal suture
neck muscle fossa
squamosal
squamosal fossa

**Figure 4** **Right lateral views of the skull of *Kentriodon sugawarai* sp. nov., holotype, NMHF 999.** (A) Photo. (B) Corresponding line drawing with anatomical interpretations. Scale bar equals 100 mm.

**Formation and Age**: Although the precise locality of NMHF 999 is at present uncertain, and the exact horizon from which NMHF 999 was collected is also unclear, the siltstone matrix adhering to NMHF 999 has produced a diatom flora that includes *Denticulopsis praelauta* (*Oishi et al., 1999*). Consequently, NMHF 999 should come from the middle or upper portions of the Kadonosawa Formation, because the Shikonai Siltstone Member of this formation is dominated by silts and very fine sandstones. The Shikonai Siltstone Member of the Kadonosawa Formation is widely distributed in Ninohe City, including the

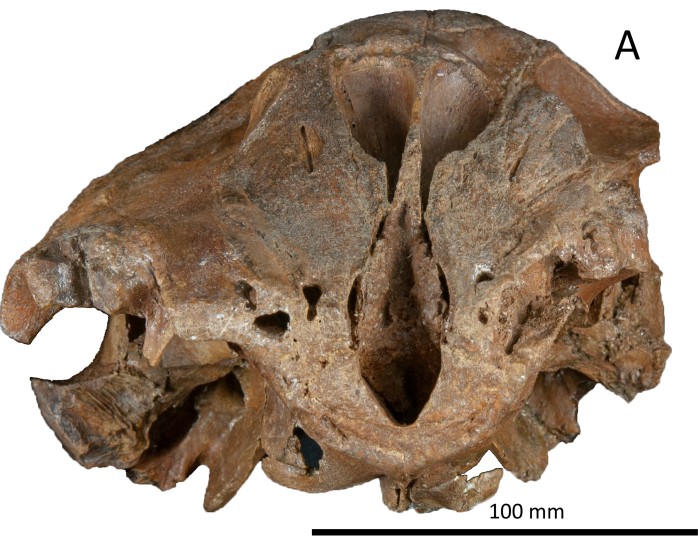

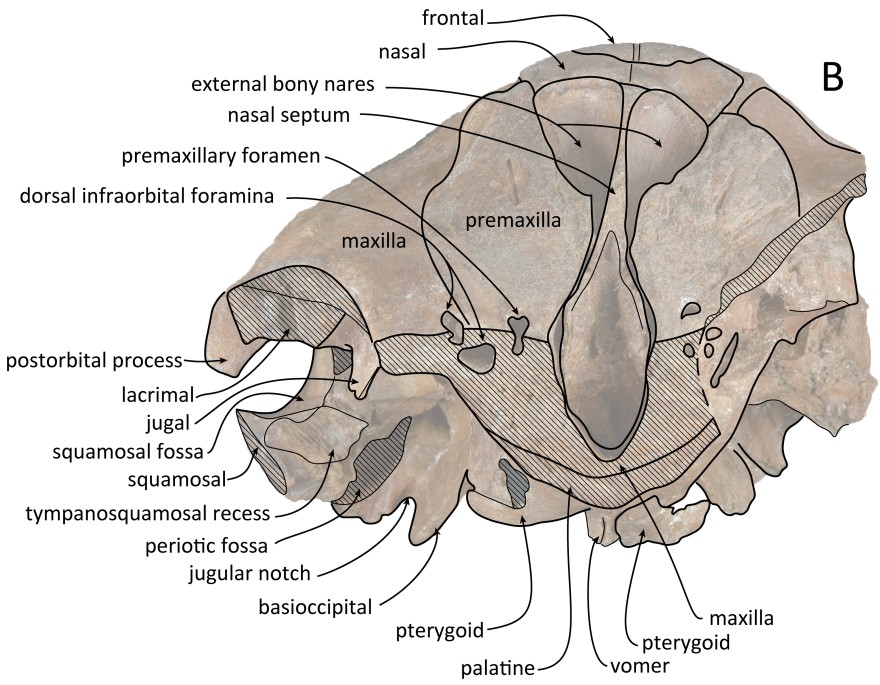

**Figure 5** **Anterior views of the skull of *Kentriodon sugawarai* sp. nov., holotype, NMHF 999.** (A) Photo. (B) Corresponding line drawing with anatomical interpretations. Scale bar equals 100 mm.

provenance area of NMHF 999. The siltstone layers of the Shirikonai Siltstone Member of the Kadonosawa Formation have produced a rich diatom assemblage, which has been referred to the *Denticulopsis praelauta* Zone (NPD 3B) (*TuZino & Yanagisawa, 2017*; *Tuzino et al., 2018*). The range in age of this zone spans chronostratigraphically between 16.3 and 15.9 Ma (*Yanagisawa & Akiba, 1998*), latest Burdigalian to earliest Langhian,

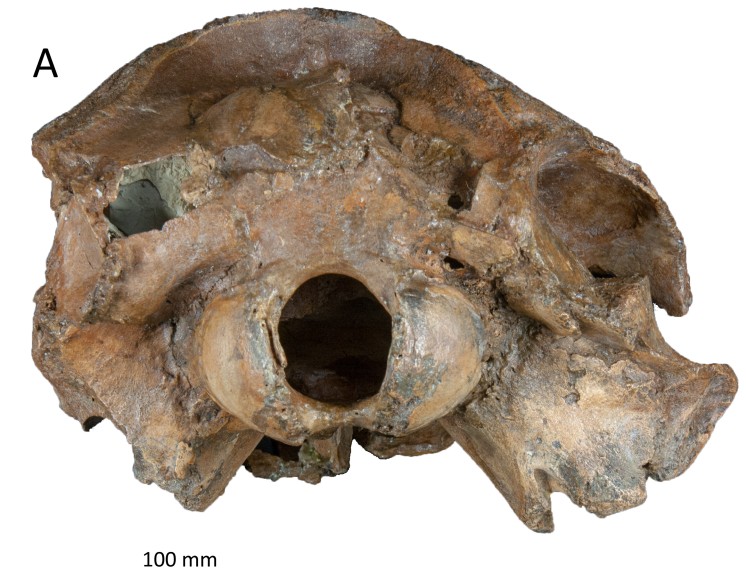

100 mm

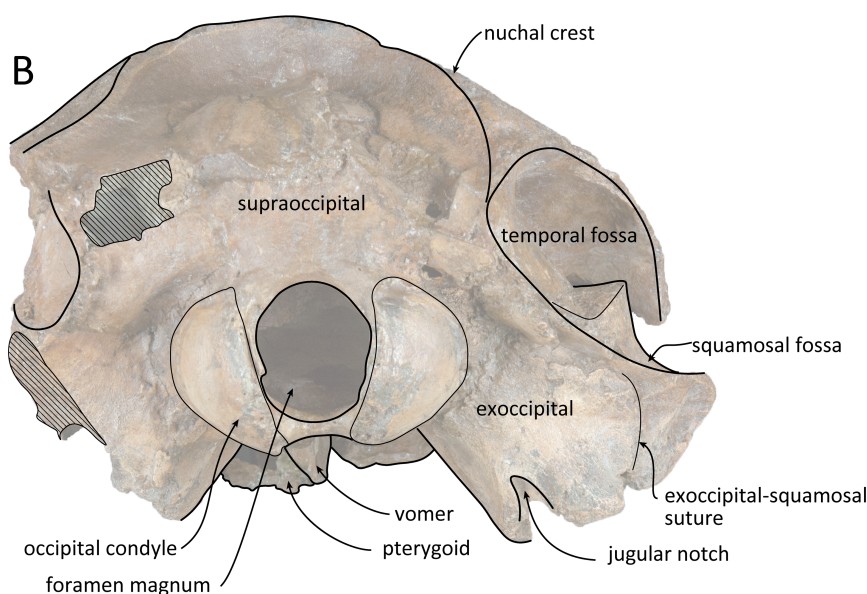

**Figure 6** **Posterior views of the skull of *Kentriodon sugawarai* sp. nov., holotype, NMHF 999.** (A) Photo. (B) Corresponding line drawing with anatomical interpretations. Scale bar equals 100 mm.

latest early to earliest middle Miocene. The main part of the Kadonosawa Formation has yielded abundant molluskan fossils (*Chinzei, 1966*), as well as a tooth of *Desmostylus* (*Oishi & Kawakami, 1984*). Based on ostracods (*Irizuki & Matsubara, 1994*) and benthic foraminifera (*Kamemaru, Matsubara & Irizuki, 1995*), the depositional environment of the Shikonai Siltstone Member of the Kadonosawa Formation conforms to sublittoral to bathyal settings.

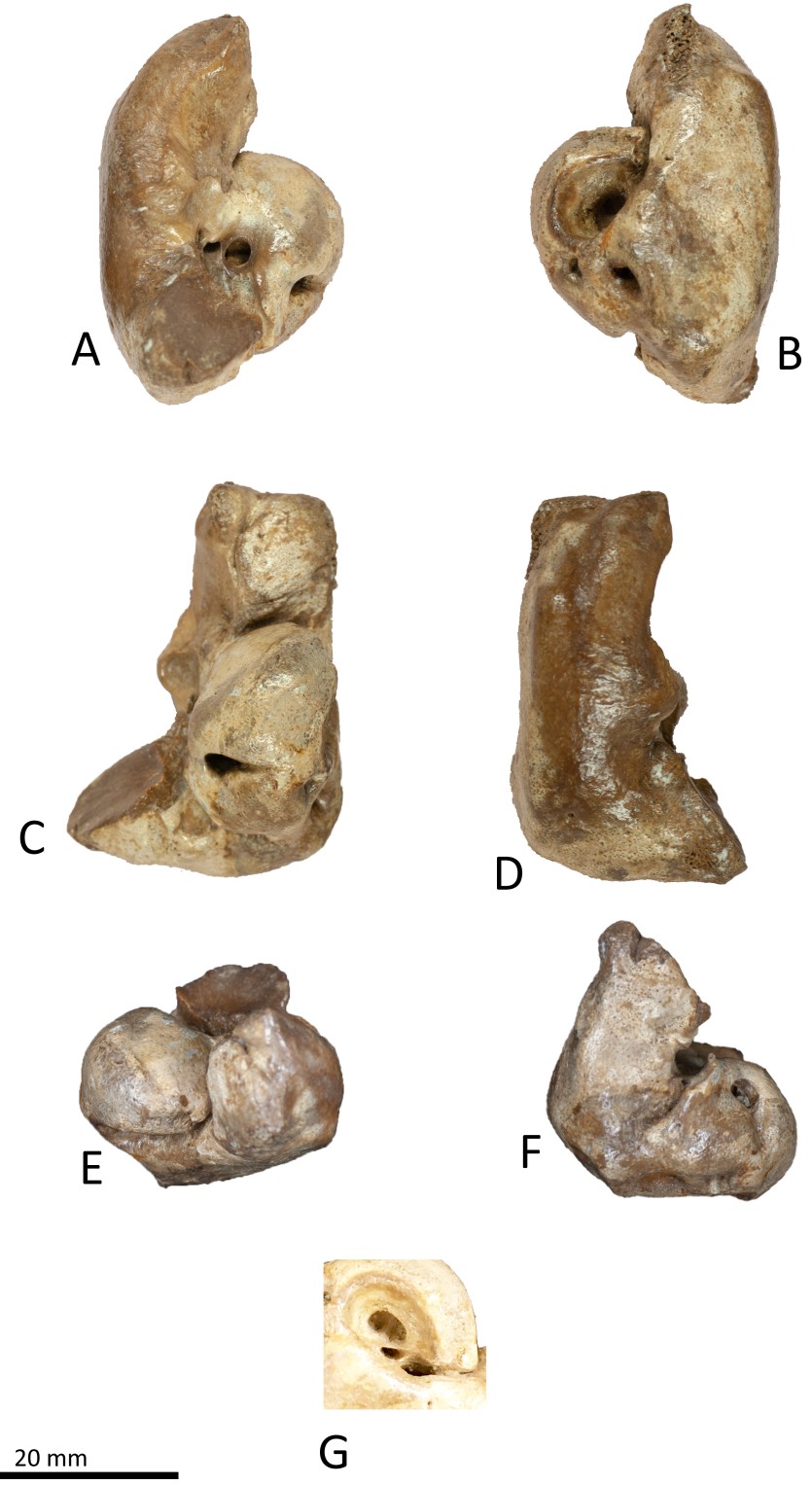

**Figure 7** **Right periotic of *Kentriodon sugawarai* sp. nov., holotype, NMHF 999.** (A) Ventral view. (B) Dorsal view. (C) Medial view. (D) Lateral view. (E) Anterior view. (F) Posterior view. (G) Detail view of the internal acoustic meatus. Scale bar equals 20 mm.

20 mm

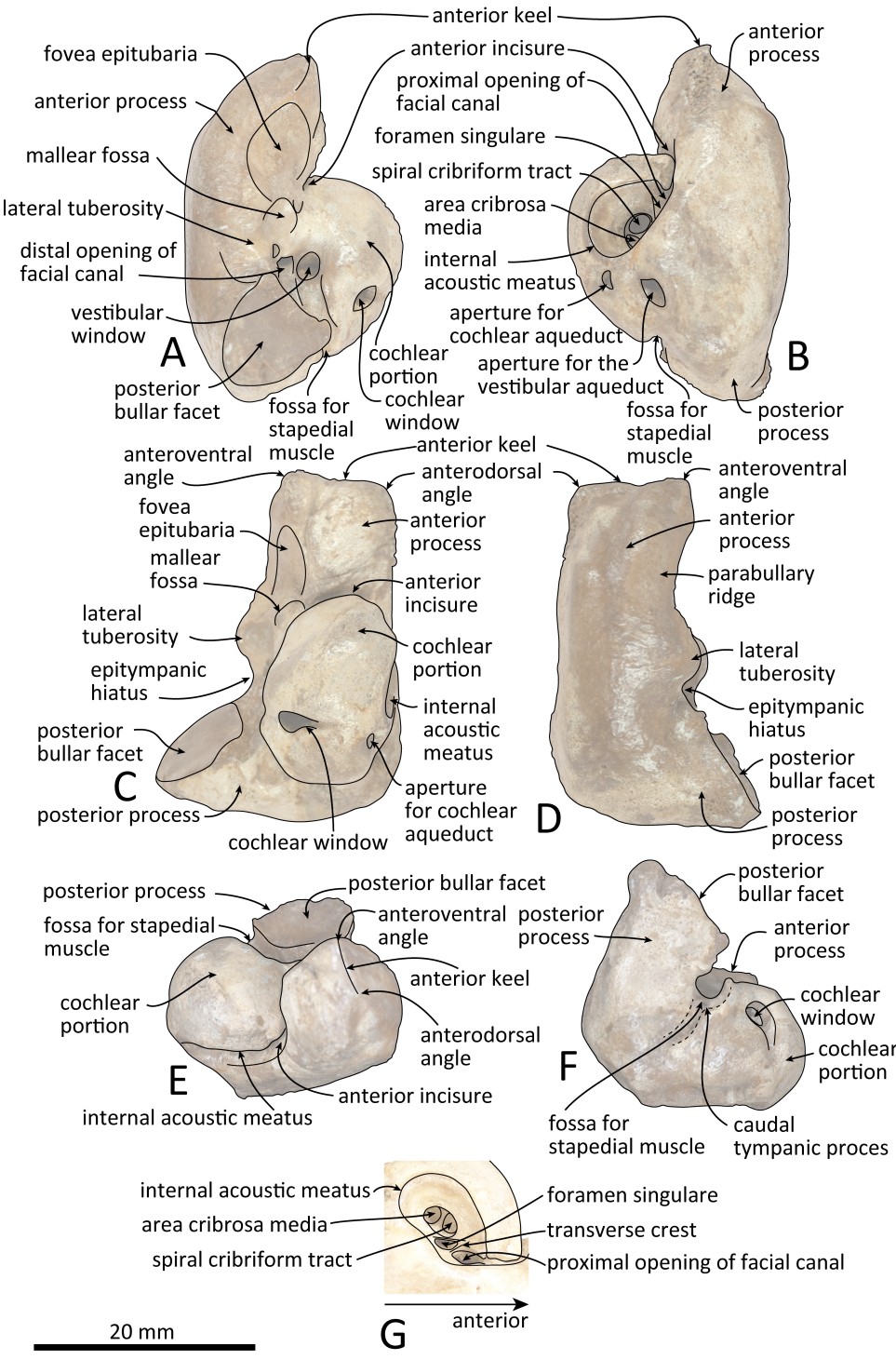

**Figure 8  Line drawings of the right periotic of *Kentriodon sugawarai* sp. nov., holotype, NMHF 999, with anatomical interpretations.** (A) Ventral view. (B) Dorsal view. (C) Medial view. (D) Lateral view. (E) Anterior view. (F) Posterior view. (G) Detail view of the internal acoustic meatus. Scale bar equals 20 mm.

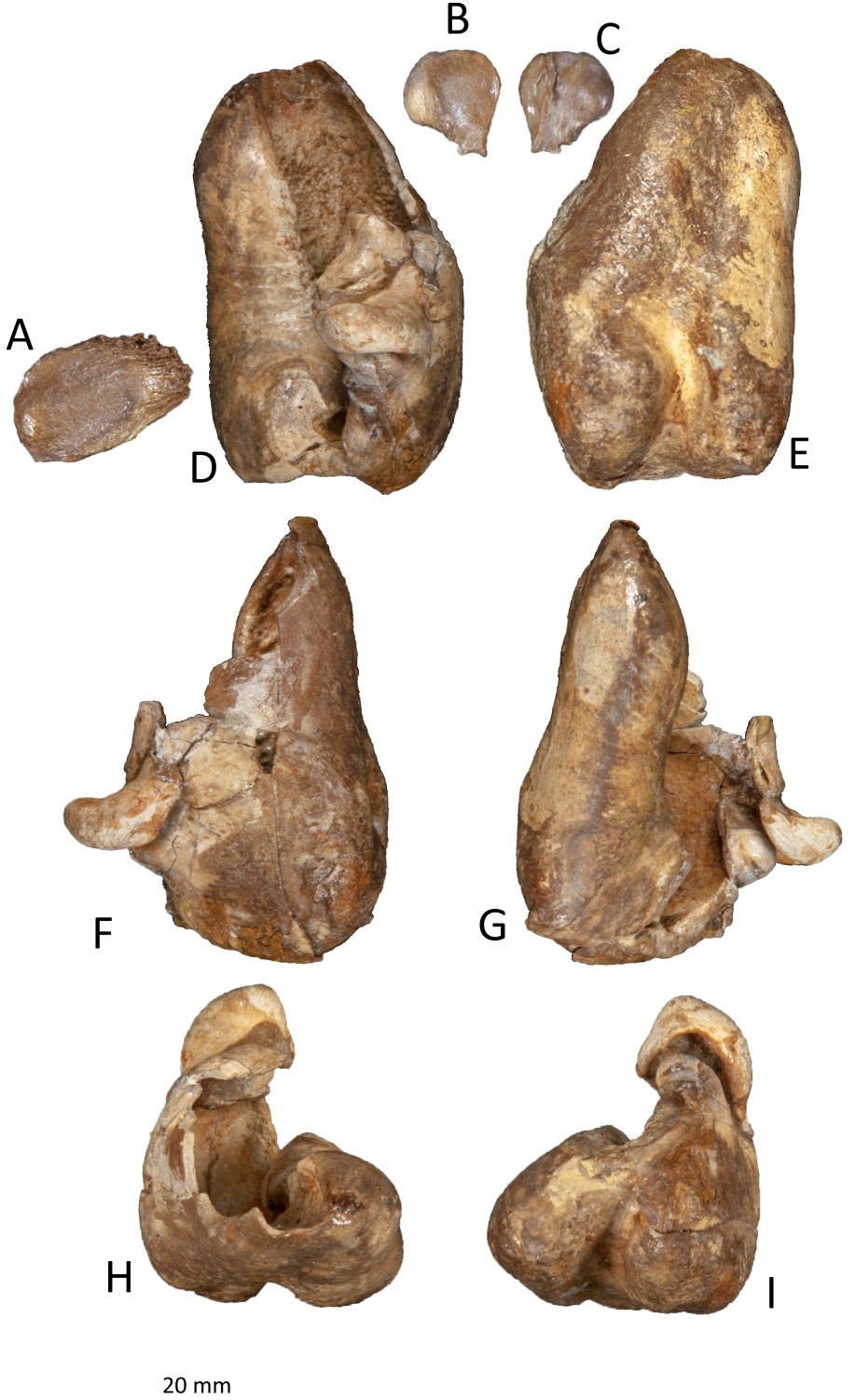

**Figure 9** **Right tympanic bulla of *Kentriodon sugawarai* sp. nov., hototype, NMHF 999.** (A) Dorsal view of the posterior process of the tympanic bulla. (B–C) accessory ossicle. (B) Dorsal view. (C) Ventral view. (D–I) Left tympanic bulla. (D) Dorsal view. (E) Ventral view. (F) Lateral view. (G) Medial view. (H) Anterior view. (I) Posterior view. Scale bar equals 20 mm.

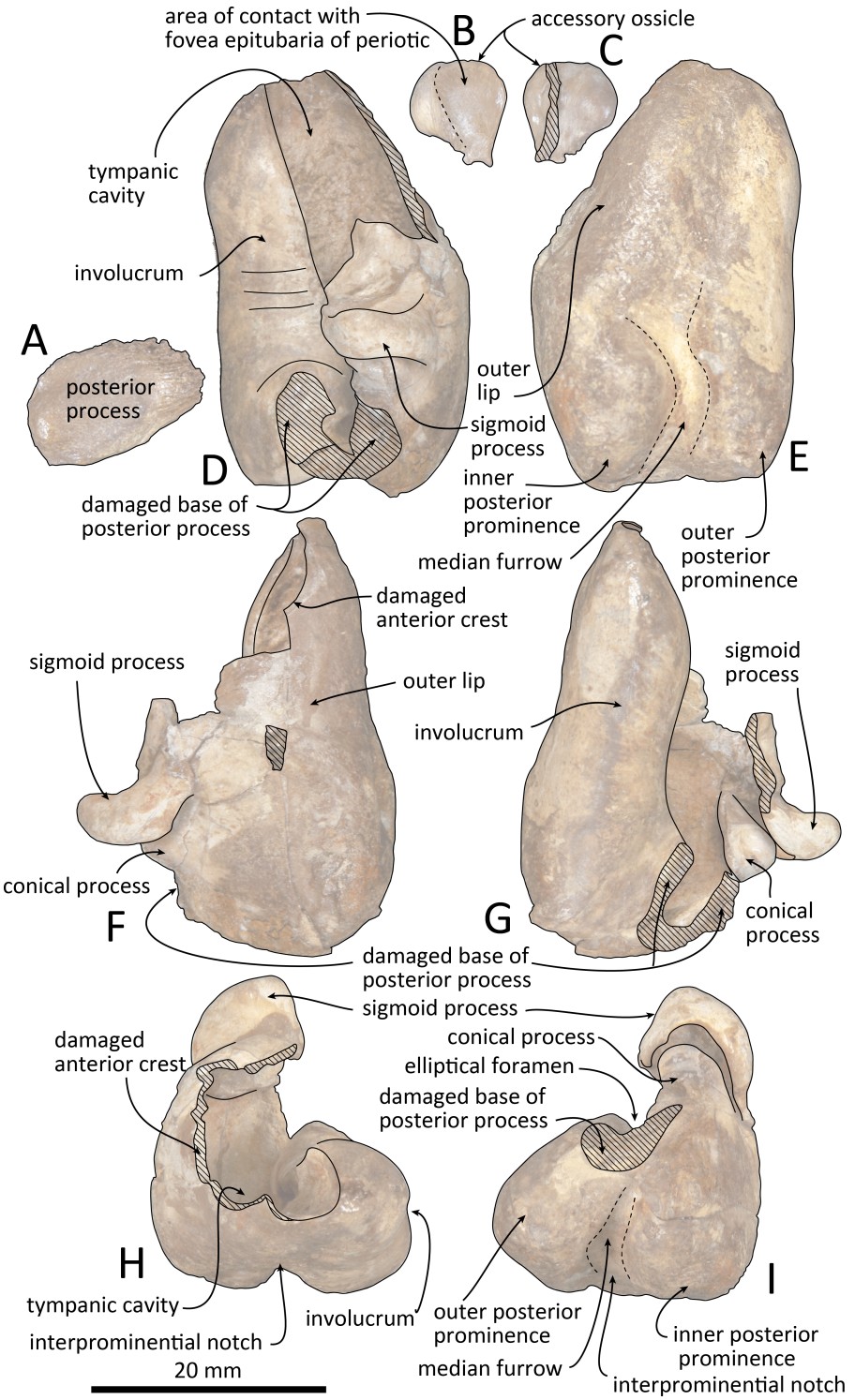

**Figure 10  Line drawings of the right tympanic bulla of *Kentriodon sugawarai* sp. nov., holotype, NMHF 999.** (A) Dorsal view of the posterior process of the tympanic bulla. (B–C) Accessory ossicle. (B) Dorsal view. (C) Ventral view. (D–I) Left tympanic bulla. (D) Dorsal view. (E) Ventral view. (F) Lateral view. (G) Medial view. (H) Anterior view. (I) Posterior view. Scale bar equals 20 mm.

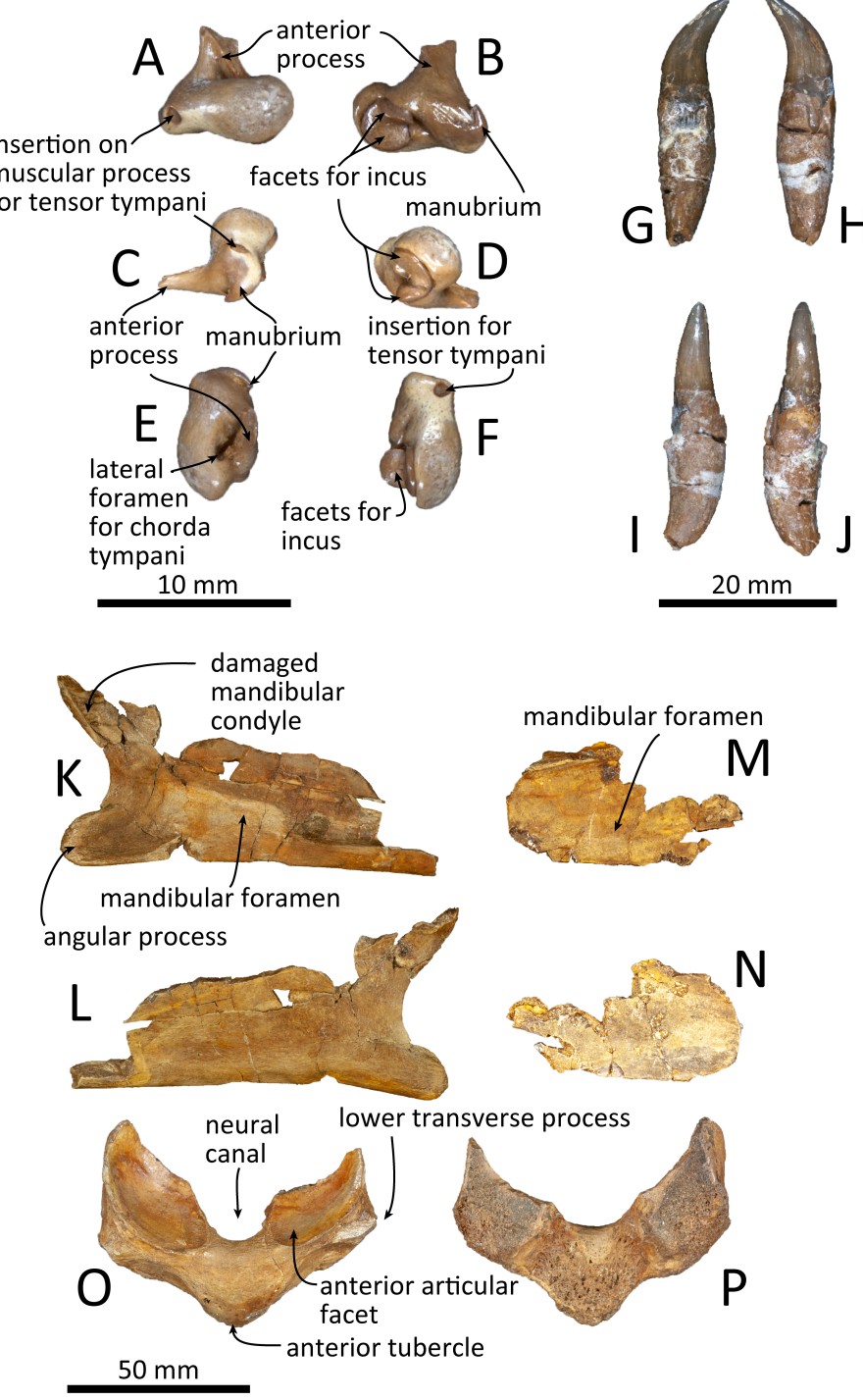

**Figure 11 Malleus, tooth, mandible and vertebra of *Kentriodon sugawarai* sp. nov., holotype, NMHF 999, with anatomical interpretations.** (A–F) Right malleus. (A) Ventral view. (B) Dorsal view. (C) Medial view. (D) Lateral view. (E) Anterior view. (F) Posterior view. Scale bar equals 10 mm. (G–J) Probable upper tooth. (G) Distal view. (H) Mesial view. (I) Lingual view. (J) Labial view. Scale bar equals 10 mm. (K–L) Ascending ramus of the left mandible. (K) Medial view. (L) Lateral view. (M–N) horizontal ramus of the right mandible. (M) Lingual view. (N) Labial view. (O–P) Ventral half of the atlas. (O) Anterior view. (P) Posterior view. Scale bar equals 50 mm.

**Table 1** Measurements (in mm) for the skull and tympanoperiotic of *Kentriodon sugawarai* sp. nov., holotype, NMHF 999.

| Dimension | Measurement |
| --- | --- |
| **Skull** | |
| Condylobasal length, from tip of rostrum to hindmost margin of occipital condyles | 186.2+ |
| Length of rostrum, from tip to line across hindmost limits of antorbital notches | 15.1+ |
| Width of rostrum at base, along line across hindmost limits of antorbital notches | 107.2e |
| Maximum length of frontal at the vertex | 19.3 |
| Width of the foramen magnum | 21.8 |
| Width of premaxillae at posterior extremity | 61.4 |
| Width of nasal bones | 53.4 |
| Distance from tip of rostrum to external nares (to mesial end of anterior margin of right naris) | 56.0+ |
| Distance from tip of rostrum to internal nares (to mesial end of posterior margin of right pterygoid) | 56.8+ |
| Greatest preorbital width | 181.2e |
| Greatest postorbital width | 215.0e |
| Least supraorbital width | 179.2e |
| Greatest width of external nares | 41.8 |
| Greatest width across zygomatic processes of squamosals | 161.0+ |
| Greatest width of premaxillae | 101.4e |
| Greatest parietal width | 138.0+ |
| Greatest length of left temporal fossa, measured to external margin of temporal crest | 110 |
| Greatest width of left temporal fossa perpendicular to greatest length | 54.5 |
| Major diameter of left temporal fossa proper | 41.6 |
| Minor diameter of left temporal fossa proper | 48.1+ |
| Distance from foremost end of junction between nasals to hindmost point of margin of nuchal crest | 34.6 |
| Length of left orbit-from apex of preorbital process of frontal to apex of postorbital process | 52.4 |
| Length of antorbital process of left lacrimal | 18+ |
| Greatest width of internal nares | 71.4e |
| Greatest length of left pterygoid | 47.6 |
| Width across occipital condyles | 70.5 |
| **Periotic** | |
| Total length | 31.7 |
| Length of anterior process | 17.8 |
| Width at pars cochlearis | 13.2 |
| Length of posterior bullar facet | 10.5 |
| Width of posterior bullar facet | 9.9 |

**Table 1** (*continued*)

| Dimension | Measurement |
|---|---|
| Length of pars cochlearis, from anterior to posterior margin | 17.2 |
| **Tympanic Bulla** | |
| Total length without posterior process as preserved | 37.4 |
| Total width as preserved | 21.8 |
| Width of inner posterior prominence | 9.5 |
| **Atlas** | |
| Width of atlas | 69.1 |
| Length of atlas | 62.9+ |
| Greatest width of facet for occipital condyle | 24.9 |

**Notes.**

Abbreviations: e, estimate; +, not complete.

# DESCRIPTION

## Cranium

The cranium lacks most of the rostrum (Fig. 2), and the left orbit and parts of the left and right squamosals are also missing. In ventral view, the cranium has been dorsoventrally crushed in the area of the choanae. The choanae are cracked and not connected with the bony nares, anteriorly depressed by secondary deformation (Fig. 3). In dorsal view, the nasals and the premaxillae are almost symmetrical, while the midline of the occipital condyles is slightly skewed to the right. The cranium underwent some degree of oblique deformation, from the upper right to the lower left and the dorsal part of the cranium might fall left by this deformation. In lateral view, the temporal fossa is anteroposteriorly long and dorsoventrally high. The vertex is low and flat, being formed by the frontals and the nasals.

## Premaxilla

The premaxillae are symmetrical. Most of the rostral portion of premaxillae is broken away. The broken section is just anterior to the antorbital notch. The premaxillary foramen is just posterior to the broken section, and at the same level as the antorbital notch. Anteromedial to the premaxillary foramen, the anteromedial sulcus and the prenarial triangle are not preserved. No posteromedial sulcus of the premaxilla was observed. The lateral margin of each side of the premaxilla is also broken, and only a recognizable premaxillary surface on right side of the maxilla remains. Anterior to the bony nares, the premaxillae are dorsoventrally thin and flat. The posterolateral sulcus cannot be recognized, and the premaxillary sac fossa is weakly depressed. In lateral view, the ascending process of the premaxilla forms an angle of about 20° from the anteroposterior axis of the cranium (Fig. 4). The knob-like posterior end of the premaxilla contacts the anterolateral corner of the nasal at a level slightly lower than the vertex of the cranium.

## Maxilla

The left maxilla is broken laterally, but the right antorbital notch is preserved. The maxillary-palatine suture and the cross-section of the infraorbital canal are observed along the broken section (Fig. 5). The maxilla is generally flat transversely in the antorbital

region, and there is no indication that any maxillary crest was present. The lateral margin of the maxilla is flat in its orbital area, whereas it is slightly concave dorsally and posteriorly to the orbit. There are three anterior and one posterior dorsal infraorbital foramina on the right maxilla (Fig. 2). The anteriormost of these foramina is located just beside the maxilla-premaxilla suture, anteromedial to the antorbital notch; other small anterior dorsal infraorbital foramina are located at the level of the orbit. The posterior dorsal infraorbital foramen is the largest, and its opening is located on the ascending process at the anteroposterior level of the corresponding nasal. The maxilla rises towards the vertex gently at the ascending process but steeply along the lateral face of the vertex. Although the vertex is low, the maxilla faces laterally just lateral to the nasal and makes a deep fossa on each side, defined medially by the nasal and posteriorly by the nuchal crest. In dorsal view, the posterior and lateral margins of the right maxilla are semicircular, and the posteromedial margin contacts the supraoccipital. In lateral view, the maxilla forms a thin plate and covers the frontal dorsally in orbital area. It gradually thickens anteriorly at the antorbital process. In ventral view, the right ventral infraorbital foramen is preserved, while the lateral edge of the left ventral infraorbital foramen is broken away. The dorsolateral edge of the opening of the ventral infraorbital foramen is formed by the maxilla and the lacrimal or the jugal.

### Palatine and pterygoid

The posterior halves of both the palatines are preserved. The palatine-maxilla suture is visible from the anterior side through the broken transverse section of the rostrum base (Fig. 5). The palatine contacts the maxilla dorsally and is dorsoventrally thinner than the maxilla. The left and right palatines are not separated medially at the level of the transverse section, just anterior to the antorbital notch. The left side of the palatine is broken laterally, but the parasagittal section of the left infraorbital canal is observed (Fig. 3). The palatine-maxilla suture is not clear in the parasagittal section, and the ventromedial edge of the ventral infraorbital foramen is uncertain.

The pterygoids are well preserved, including both the lateral and ventral laminae of the pterygoid and the pterygoid hamuli. The anterior tip of the pterygoid is located slightly posteriorly to the level of the antorbital notch. The pterygoid sinus is ventrally covered by the pterygoid. The anterior edge of the pterygoid sinus is at the level of the antorbital notch. The palatal surface of the pterygoid is flat and ventrally convex. The sagittal portion of the two pterygoids do not contact each other medially in their posterior portion. The pterygoid hamulus is short and preserving the hamular crest. The hamular crests of the left and right pterygoids diverge posteriorly in ventral view, and extend to the posterolateral–most of the lateral and ventral laminae of the pterygoid, just posterior to the infratemporal crest of the frontal. The medial lamina forms the anterolateral wall of the internal nares. Although the pterygoid-basioccipital suture is not preserved clearly, the posterior lamina overlaps the basioccipital crest. It forms the pharyngeal crest and covers the alisphenoid ventrally.

### Vomer

The vomer is visible dorsally. When viewed from the anterior transverse section (Fig. 5), the premaxilla does not roof the vomer dorsally. The mesorostal groove is widely open as a

U-shaped groove, starting 11 mm anterior to the anterior edge of the bony nares. In ventral view, the vomer does not seem to make contact with its posterior part due to deformation. It twists left and ventral to the posterior part (Fig. 5). The posterior part of the vomer is posteriorly overhung beneath the basisphenoid.

### Presphenoid

The nasal septum is narrow transversely and straight dorsally. It is as high as the level of the nasal process of the premaxilla and posteriorly contacts both the nasals. Because of this, the cribriform plate cannot be distinguished in dorsal view.

### Nasal

The shape of the nasal in dorsal view is subtriangular, and its transverse width is greater than its anteroposterior length. The left and right nasals are symmetrical. The nasals are slightly wider than the widest part of the bony nares. It preserves a posteriorly directed tip, slightly wider than its anterior margin. The lateral margin contacts the posterior end of the premaxilla anterolaterally and contacts the maxilla laterally. The nasals also contact the frontals posteromedially. The dorsal surface of the nasal is flat except for the slightly concave anterolateral part. There is no indication of an internasal fossa, and the internasal suture is a shallow trough. In dorsal view, the anterior border of the nasal is slightly retracted posteriorly from the bony nares.

### Frontal

The frontal is only exposed dorsally at the vertex. In dorsal view, the frontal is separated from the maxilla by the nasal. The dorsal exposure of the joined frontals on the vertex is elliptical, and each frontal contacts the corresponding nasal anterolaterally and the supraoccipital posteriorly, but not contacting with the maxilla laterally. The frontal at the vertex is slightly higher than the nasal, being the highest point of the cranium. In lateral view, the orbit is markedly concave and low, in line with the lateral edge of the posterior part of the rostrum (Fig. 4). In lateral view, the preorbital process is thick anteriorly. While the anteriormost part of the frontal is broken, the frontal-lacrimal suture is not clear. Although being distally broken, the postorbital process is somewhat transversely narrow and triangular at the base, being directed posteroventrally or ventrally. Both the fossae for the preorbital and the postorbital lobes of the pterygoid sinus are shallow or absent. The frontal groove is deep medially, extends anterolaterally beyond the level of the ventral infraorbital foramen. The infratemporal crest is curved, distinct just lateral to the optic canal.

### Lacrimojugal complex

The left antorbital process is broken, and the right antorbital process is also broken anteriorly and laterally, so that the shape and the anteriormost portion of the antorbital process is not clear. Although both the lacrimal-maxilla suture and the lacrimal-jugal suture are not clear in dorsal view, the lacrimal-maxilla suture might be observable in anterior view along the broken section of the antorbital process. Here the lacrimal appears as thicker than the maxilla. The right jugal contacts the lacrimal exactly at the posterior

end of the antorbital notch. The main part of the right jugal is broken, only preserved as a short narrow base of the maxillary process.

## Squamosal

The right squamosal is almost missing, while the left squamosal only preserves the postglenoid process, with the zygomatic process being lost. The zygomatic process of the squamosal is likely to be long because the postorbital process is relatively far from the postglenoid process. The squamosal fossa is incomplete anteriorly, but it is shallow and somehow longer anteroposteriorly and transversely wide, and it faces dorsally. The lateral margin of the fossa is visible in dorsal view. The mandibular fossa is not preserved, while the tympanosquamosal recess can be observed medially. The tympanosquamosal recess is flat and wide, and its ventral surface is wrinkled. The falciform process is not well preserved. The dorsal roof of the external auditory meatus is preserved and is distinctively narrow, but the postglenoid process just in front of the external auditory meatus is not clear because of insufficient preservation.

## Supraoccipital

The supraoccipital extends broadly in dorsal view. The nuchal crest is trapezoidal in outline, while it is medially concave at the level of the temporal fossa. It expands ventrolaterally toward the exoccipital. In posterior view, the supraoccipital is inclined anteriorly with a reduced dorsoventral height. The occipital shield is concave anterodorsally and convex posteroventrally. Anteromedially, just posterior to the nuchal crest, the supraoccipital is concave dorsally, forming a fossa whose anteriormost surface faces posteriorly. Posteriorly, the occipital shield bulges medially (Fig. 6) and it is collapsed along the right margin, most likely as a result of deformation. The supraoccipital is fused with the exoccipital along an undefined suture. There is no indication of an external occipital crest.

## Exoccipital

The exoccipital is wide in posterior view. It extends laterally from the temporal crest and is fused with the basioccipital ventrally. The temporal crest overhangs the exoccipital posterolaterally, and extends nearly to the posterior most level of the cranium, not taking account of the occipital condyles. The occipital condyle is prominent, and the condylar neck is well developed, while there is no indication of a dorsal condyloid fossa. The foramen magnum is almost circular, being only slightly higher than wide. In posterior view, the jugular notch is deep and narrow. The paroccipital process is wide. The hypoglossal foramen opens in the jugular notch. The paroccipital concavity is deep.

## Basioccipital

The basioccipital basin is broad and strongly concave in posterior view. The basioccipital crests face ventromedially. In ventral view, the basioccipital crest width is transversely narrow. Anteriorly, the basioccipital contacts with the posterior lamina of the pterygoid. Its posterior margin is rounded. Medial to the crest, the ventral surface of the basioccipital is flat. The muscular tubercle is not developed in the basioccipital basin.

## Periotic

The right periotic is preserved (Figs. 7 and 8). In dorsal and ventral views, the apex of the anterior process is mediolaterally flattened. Both anteroventral and anterodorsal angles are respectively tapered and directed anteriorly, reaching the same level anteriorly and separated by the anterior keel. The anteroposterior length of the anterior process is nearly the same as that of the pars cochlearis. The anterior incisure is deep, and it separates the anterior process from the pars cochlearis. In lateral view, the ventral surface of the parabullary ridge is concave. There is a flat surface anterior to the fovea epitubaria, which is circular and about two mm long, which might correspond to a very shallow anterior bullar facet. The fovea epitubaria is broad, and it receives with no fusion the accessory ossicle of the tympanic bulla. There is a fossa posteromedial to the fovea epitubaria and anteromedial to the mallear fossa. It receives the tubercle of the malleus. The mallear fossa is rounded and faces ventrally rather than medially. The lateral tuberosity is bulbous lateral to the mallear fossa. The epitympanic hiatus is concave just posterior to the lateral tuberosity. The hiatus just located posteromedially to the facial canal. The vestibular window is rounded and slightly larger than the opening for the facial canal.

The medial outline of the pars cochlearis is rounded and compressed dorsoventrally. The cochlear window opens on the posterior wall of the pars cochlearis. The aperture for the cochlear aqueduct opens dorsally and is located close to the vestibular aqueduct at the same transverse level as the medial edge of the internal acoustic meatus. The aperture for the vestibular aqueduct is two times larger than the aperture for the cochlear aqueduct and located slightly posteriorly to the aperture for the cochlear aqueduct. The internal acoustic meatus is large and funnel-shaped, with an anterolateral–posteromedial axis. The anteriormost tip of the internal acoustic meatus extends in the anterior incisure. The foramen singulare is located closer to the proximal opening of the facial canal than to the spiral cribriform tract, and it is separated by partitions from the proximal opening of the facial canal and the spiral cribriform tract. The proximal opening of the facial canal is located slightly anterior to the spiral cribriform tract. The area cribrosa media has almost the same size as the spiral cribriform tract. The posterior process extends for a short distance anteroventrally, while its posterior edge is directed ventrally. In lateral view, the posterior and dorsal faces of the posterior process draw a blunt right angle. The posterior bullar facet is smooth and faces anteroventrally.

## Tympanic bulla

The right tympanic bulla only lacks the anterodorsal crest (Figs. 9 and 10). The accessory ossicle is preserved but detached from the tympanic bulla. The tympanic bulla is narrow and long in lateral view, and its ventral margin is slightly concave in lateral view. The lateral furrow is absent or very shallow. The disarticulated accessory ossicle is rounded, 8.3 mm in width. It originally occupied the fovea epitubaria of the periotic. In ventral view, there is an anteroposterior linear fracture surface on the accessory ossicle, for the attachment to the outer lip of the tympanic. The involucrum tapers anteriorly, and the anterior spine is absent. The dorsal and ventral margins of the involucrum are parallel and the dorsal margin is excavated just anterior to the posterior process. The ventromedial keel on the

involucrum is not clearly defined. The interprominential notch is shallow and is followed anteriorly by the median furrow. The median furrow is shallow. It widens anteriorly and points laterally. In dorsal view, the sigmoid process is large and rounded, and it partly covers the conical process. The posterior edge of the sigmoid process is thick. The conical process is dorsally high. The inner and outer posterior prominences extend posteriorly to the same level, and they are almost equal in transverse width. The posterior process, which is broken at its base, is rounded in outline and thick. The facet for the posterior process of the periotic is smooth. The elliptical foramen can be observed between the outer and inner posterior pedicles.

### Malleus

The malleus is isolated from the periotic (Figs. 11A–11F). The head is high anteromedially. The ventral margin of the tubercule is concave. The manubrium of the malleus forms a hook-like process at the medial margin, which directs anteriorly. The insertion for the tendon of the m. tensor tympani opens ventrally. The processus muscularis is small.

### Mandible

Both the left and right mandibles are partly preserved (Figs. 11K–11N), but the right mandible only preserves its posterior part, while the left mandible preserves its posteroventral part including the angular process and the ventral half of the mandibular condyle. The mandibular foramen is shallow mediolaterally. The left mandible does not preserve the anterior margin of the mandibular foramen. The posterior margin of the angular process is rounded, and the medial surface is concave. The mandibular condyle is located more posteriorly than the angular process and is separated from the latter by an anteriorly inward, deep, rounded curve. The medial surface of the condyle is concave and the condylar articular surface is not preserved.

### Tooth

One isolated tooth is preserved (Figs. 11G–11J). It is small and conical, at least 27 mm long and with a maximum diameter of the root of 6.3 mm. The crown surface is smooth and its apex is curved. The tooth root is also smooth and conical, and it is 1.5 times longer than the crown. The cementum of the root is just slightly thicker than the crown, and the tip of the root is recurved in a direction that is at right angle with the curve of the crown's apex.

### Vertebra

Only a fragment of the atlas is preserved (Figs. 11O, 11P). The ventral part of the bone, without the upper transverse processes, is preserved. In posterior view, the posterior articular surface is not well preserved, but it was originally not fused with the axis. In dorsal view, the anterior tubercle is short anteroposteriorly and relatively high dorsoventrally. It bears a V-shaped crest on its anterior surface, starting from the ventral most of the anterior articular facet and running to the ventral apex of the anterior tubercle.

## RESULTS OF PHYLOGENETIC ANALYSES

Our phylogenetic analysis found 256 most parsimonious trees with 3424 steps of total branch length. Each tree has a consistency index of 0.197 and a retention index of 0.564.

The 50% majority rule consensus of those trees is shown in Fig. 12, and the strict consensus tree is shown in the Fig. 13 (see also Supplemental Information). Both consensus trees show that all the species that were previously identified as kentriodontids form a monophyletic group that is positioned as the sister group to the crown Delphinoidea (i.e., Delphinidae, Phocoenidae and Monodontidae), and the clade Lipotidae + Inioidea is a sister to the clade Delphinoidea (monophyletic Kentriodontidae + crown Delphinoidea).

Delphinoidea differ from Lipotidae + Inioidea by the following 11 synapomorphies: the anterior sinus fossa is located between the anterior extremity of the pterygoid sinus and the posterior extremity of the upper tooth row (Chr. 19), the apex of the postorbital process of frontal is directed ventrally (Chr. 61), the width of the premaxillae at the antorbital notches is moderate (Chr. 67), the apex of the anterior process of the periotic is thickened by the prominent dorsal tubercle that gives to this apex a rectangular section on the plane of the body of the periotic (Chr. 239), the contact of the anterior process of the petrosal with a portion of the ectotympanic bulla anterior to the accessory ossicle is absent (Chr. 249), the periotic articulates with the squamosal along the hiatus epitympanicus and adjacent regions on the posterior process (Chr. 286), the posterior process of the periotic is long (Chr. 292), lateral furrow of the tympanic bulla is present as a shallow groove (Chr. 303), the ventral margin of the tympanic bulla is concave in lateral view (Chr. 307), the basihyal and the thyrohyal are fused (Chr. 322), and the roof of the neural canal of the atlas is straight (Chr. 327).

The monophyly of Kentriodontidae is supported by the following 14 synapomorphies: premaxillae are compressed mediolaterally at anterior of the rostrum (Chr. 3), the mesorostral groove is constricted posteriorly, anterior to the nares and behind the level of the antorbital notch, then rapidly diverging anteriorly (Chr. 7), anterior edge of the supraorbital process is oriented anterolaterally, forming an angle with the longitudinal axis of the skull between 35° and 60° (Chr. 50), the dorsolateral edge of internal opening of the infraorbital foramen is formed by the lacrimal or the jugal (Chr. 58), the infratemporal crest forms well-defined curved ridge on the posterior edge of sulcus for the optic nerve (Chr. 64), the premaxillary foramen is located medially (Chr. 72), the alisphenoid is broadly exposed laterally in the temporal fossa (Chr. 160), suture between the joined palatines and the joined maxillae is straight transversely or bowed anteriorly (Chr. 179), the external auditory meatus is wide (Chr. 225), angle formed by basioccipital crests as approximately 15–40° in ventral view (Chr. 229), the hypoglossal foramen is separated from the jugular foramen or the jugular notch by thick bone (Chr. 231), most convex part of the pars cochlearis is on the ventrolateral surface (Chr. 283), the basihyal is fused with the thyrohyal (Chr. 332), and the lateral edge of transverse processes of lumbar vertebrae makes an angle of 45° or more relative to the parasagittal plane (Chr. 334). However, the last two characters (i.e., Chrs. 332 and 334) are not known for most kentriodontid taxa.

The monophyly of the genus *Kentriodon* was recognized by five unique characters as was mentioned in the generic diagnosis. In particular, '*Rudicetus*' *squalodontoides* was recognized as a sister taxon to *Kentriodon diusinus* and consequently bracketed among the species of *Kentriodon*. NMHF 999 was also nested in the genus *Kentriodon* and recognized as a sister taxon to the clade of *K. pernix, K. nakajimai* and *K. obscurus* (Figs. 12 and 13).

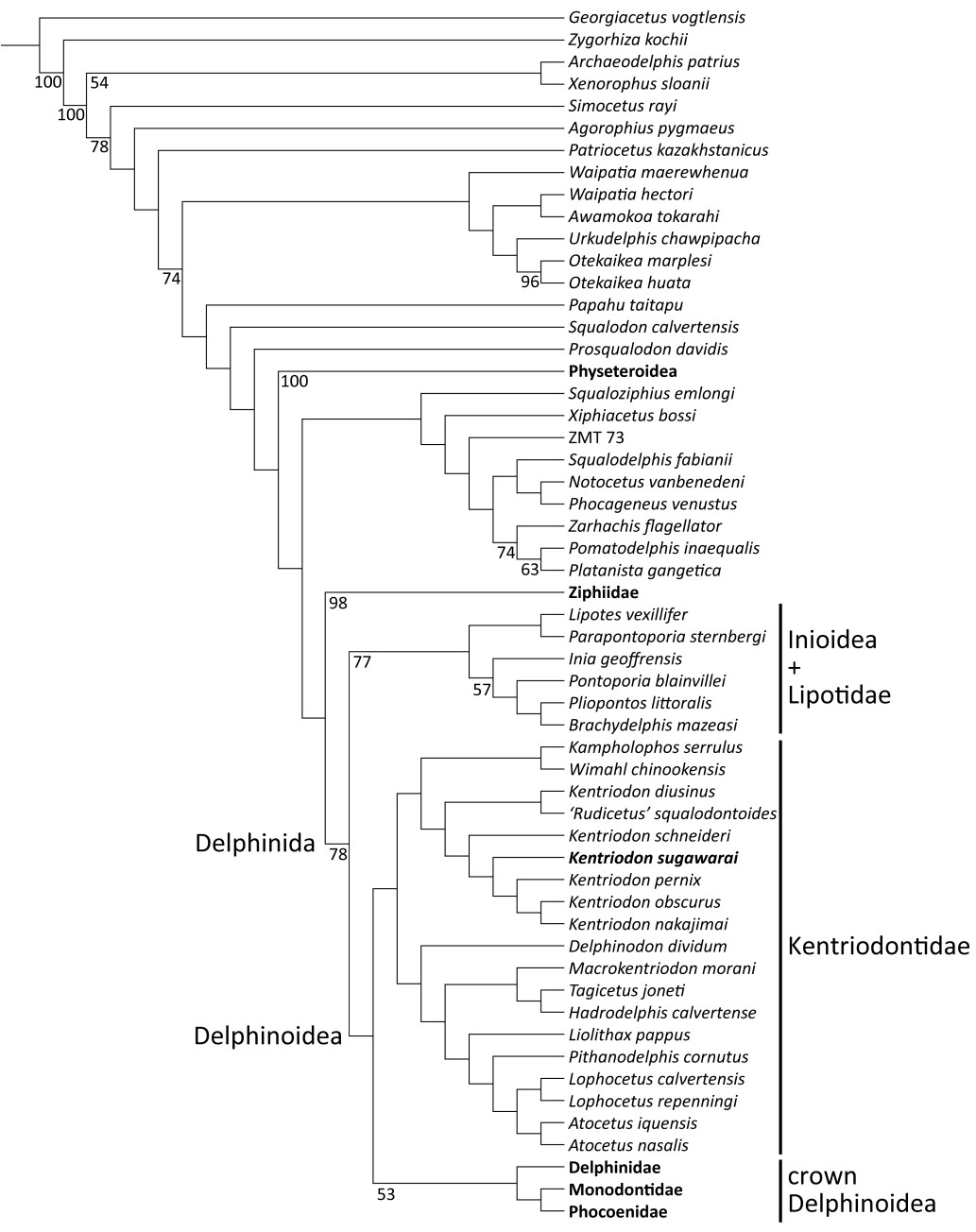

**Figure 12** **Fifty percent majority consensus tree showing the phylogenetic relationships of *Kentriodon sugawarai* sp. nov.** Fifty percent majority consensus tree resulting from 256 most parsimonious trees with tree constraint by the molecular consensus trees from *McGowen, Spaulding & Gatesy (2009)*, *McGowen et al. (2011)* and *McGowen et al. (2020)*, 3424 steps long, with the consistency index = 0.197 and the retention index = 0.564. Numbers below nodes indicate bootstrap values (1,000 replicates). The values lower than 50% were omitted. The interspecific relationships within clades Physeteroidea, Ziphiidae, Delphinidae, Phocoenidae, and Monodontidae were omitted and these groups were collapsed to families/superfamilies.

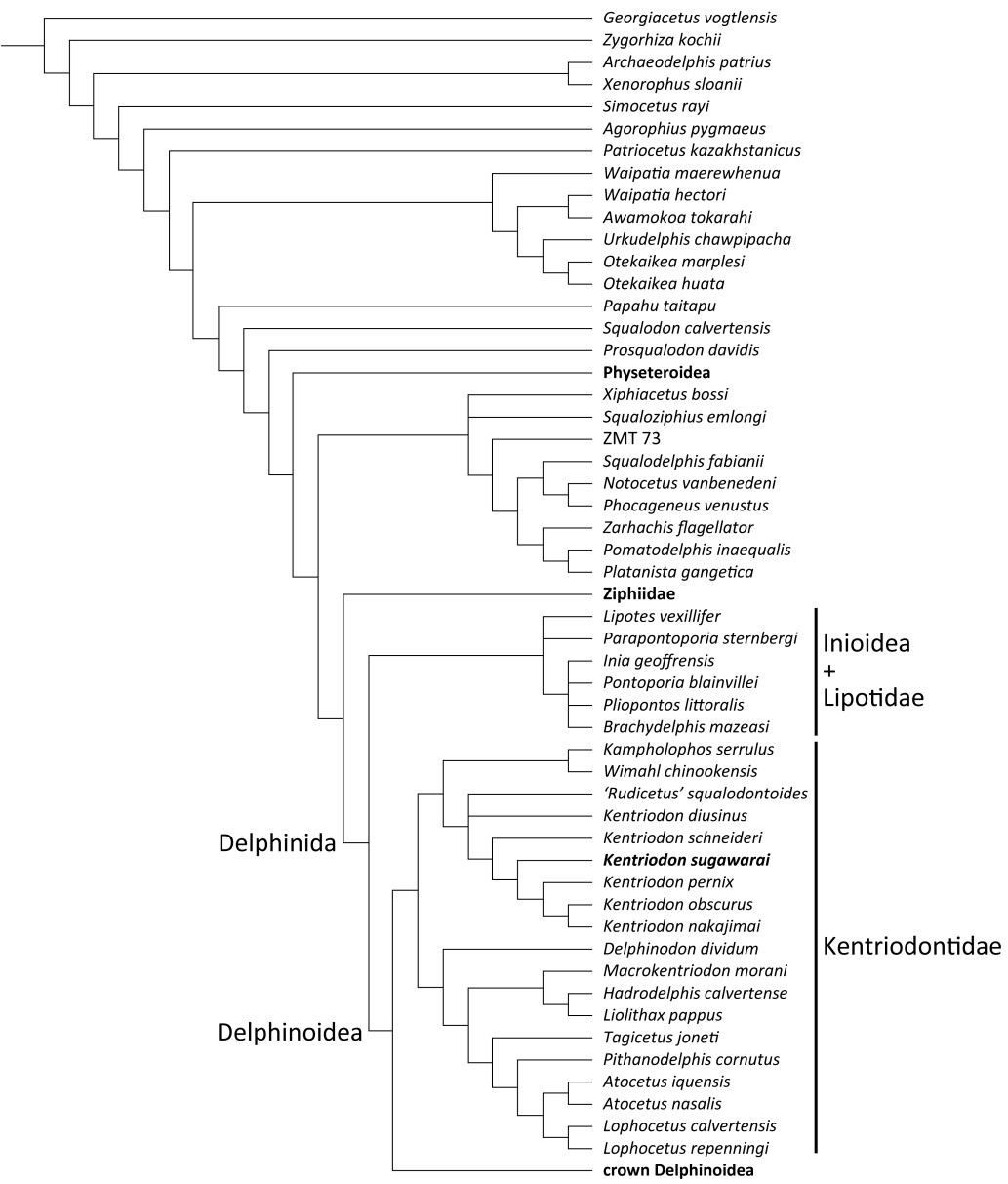

**Figure 13** **Strict consensus tree showing the phylogenetic relationships of *Kentriodon sugawarai* sp. nov.** Strict consensus tree resulting from 256 most parsimonious trees with trees constraint by the molecular consensus trees from *McGowen, Spaulding & Gatesy (2009)*, *McGowen et al. (2011)* and *McGowen et al. (2020)*, 3424 steps long, with the consistency index = 0.197 and the retention index = 0.564. The interspecific relationships within clades Physeteroidea, Ziphiidae, Delphinidae, Phocoenidae, and Monodontidae were omitted and these groups were collapsed to families/superfamilies.

## Comparison

As in most kentriodontids, the premaxillae and nasals of NMHF 999 are symmetrical. Also, the vertex of NMHF 999 is low and flat, similar to most species of *Kentriodon*. Unlike the condition in *K. hobetsu*, *K. pernix* and *K. schneideri*, the maxillae of NMHF 999 makes a deep fossae on each side and faces laterally at the vertex, intently medially by the nasal and

posteriorly by the nuchal crest. This feature is similar to *K. nakajimai* and some other genera of kentriodontids such as *Delphinodon dividum*, *Hadrodelphis calvertense*, *Liolithax pappus*, *Lophocetus calvertensis*, *Lophocetus repenningi* and *Macrokentriodon morani*. The nasal septum of NMHF 999 is high, as high as the nasal process of the premaxilla. This feature is unique to NMHF 999 and unlike other *Kentriodon* nor other genera of kentriodontids. Since the nasal septum is fragile, it is not always optimally preserved in fossils. In dorsal view, the nasal of NMHF 999 is similar to that in *K. pernix*, *D. dividum* and *Tagicetus joneti*. The condition is different from that in *K. pernix* and *D. dividum*, as the nasal of NMHF 999 posteriorly extends to the nuchal crest, and the frontal is not contacting with the maxilla laterally. However, in *T. joneti*, it also has a well−developed posterolateral projection of the nasal. The supraoccipital of NMHF 999 is concave dorsally and anteriorly, just posterior to the line of the nuchal crest at the vertex. It is similar to the condition in *K. nakajimai*, *K. schneideri*, and '*R.*' *squalodontoides*, but different from some other *Kentriodon* species, such as *K. pernix* and *K. hobetsu*. In dorsal view, the occipital shield of NMHF 999 is convex posteriorly as in *K. pernix*. Although this feature may be emphasized by deformation in NMHF 999, the same portion is straight posteriorly in most other species of *Kentriodon*. In ventral view, the tympanosquamosal recess of NMHF 999 is flat and wide, similar to that in *K. pernix* and *K. hobetsu*.

The apex of the anterior process of the periotic of NMHF 999 is directed anteriorly in dorsal views, and the anteroposterior length of the anterior process is as great as the length of the pars cochlearis. These conditions are similar to those in *H. calvertense*, *W. chinookensis*, *Liolithax kernensis* and *L. pappus*. In contrast, in *K. nakajimai*, *K. obscurus*, *K. hoepfneri* and *K. pernix*, the apex of the anterior process of the periotic is directed somewhat anteromedially, and the anterior process of the periotic is shorter than the pars cochlearis. In NMHF 999, the lateral tuberosity of the periotic is ventrally as high as that in *K. nakajimai* and *K. pernix*, but it is higher in *K. hoepfneri* and other genera of kentriodontids (i.e., *L. pappus*, *W. chinookensis*, *K. serrulus* and *Sophianacetus commenticius*). In dorsal view, the anterolateral margin of the pars cochlearis is separated from the anterior process by an anterior fissure of the facial canal in NMHF 999. This feature is also observed in *K. pernix*, *L. kernensis* and *W. chinookensis*. The interprominential notch of the tympanic bulla is shallow in NMHF 999, as in *D. dividum K. serrulus* and *W. chinookensis*, while this notch is much deeper in *A. iquensis*, *K. nakajima*, *K. pernix*, and *S. commenticius*.

Based on our phylogenetic analysis and those comparisons, identification of NMMF 999 as a distinct species within the genus *Kentriodon* is warranted. Thus, we propose the new species *Kentriodon sugawarai* sp. nov.

## DISCUSSION AND CONCLUSIONS

### Phylogenetic position of kentriodontids

Our analysis suggests that all the species of kentriodontids form a monophyletic group. Although the monophyly of the kentriodontids has been proposed in some earlier studies (e.g., *Barnes, 1978*; *Barnes, 1985*; *Muizon, 1988a*), the intergeneric and interspecific relationships therein proposed for the members of this family are both different from our

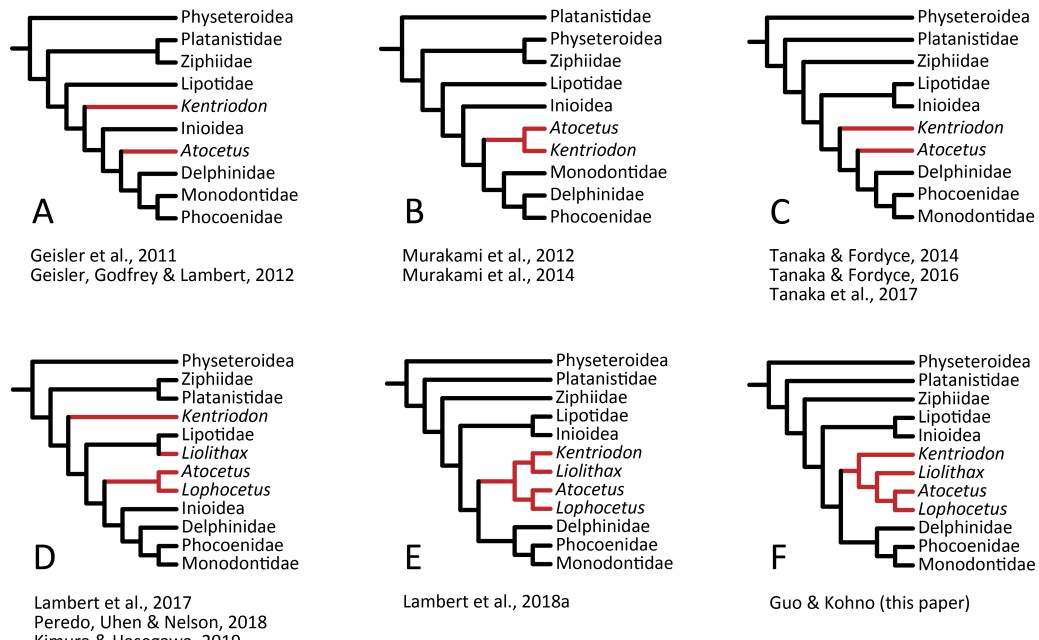

**Figure 14   Several hypotheses of the phylogeny for the Delphinida.** Topologies have been modified from previous studies, all trees are selected from unweighted and unordered. The kentriodontid lineages are colored in red. (A) Unconstrained tree from (*Geisler et al., 2011*; *Geisler, Godfrey & Lambert, 2012*). (B) Unconstrained tree from (*Murakami et al., 2012*; *Murakami et al., 2014*). (C) Molecular constraint tree from (*Tanaka & Fordyce, 2014*; *Tanaka & Fordyce, 2016*; *Tanaka et al., 2017*). (D) Molecular constraint tree from (*Lambert et al., 2017*; *Peredo, Uhen & Nelson, 2018*; *Kimura & Hasegawa, 2019*). (E) Molecular constraint tree from (*Lambert et al., 2018a*). (F) Molecular constraint tree from this paper.

results (Fig. 14). Here we suggest that kentriodontids are divided into two monophyletic subgroups (Figs. 12 and 13). The first subgroup includes *Kampholophos*, *Wimahl*, '*Rudicetus*', and *Kentriodon*, while the second subgroup includes *Delphinodon*, *Tagicetus*, *Macrokentriodon*, *Liolithax*, *Hadrodelphis*, *Pithanodelphis*, *Lophocetus*, and *Atocetus*. The 50% majority rule consensus tree (Figs. 12 and 13) shows agreements with *Lambert et al. (2017)*, *Peredo, Uhen & Nelson (2018)* and *Kimura & Hasegawa (2019)* at least as regards the former subgroup (Fig. 14). In this regard, the monophyly of the former subgroup is considered to be robust. On the other hand, other kentriodontids had been subdivided into five paraphyletic or polyphyletic groups by *Lambert et al. (2017)* and *Peredo, Uhen & Nelson (2018)*. Particularly, both of their unconstrained analyses suggested that a kentriodontid species *Liolithax* was recognized as a sister taxon to the Lipotidae, meaning that it was recognized to be more closely related to Iniidae + Pontoporiidae than other 'kentriodontids'.

Particularly, the result of our phylogenetic analysis is somewhat similar to that obtained by *Tanaka et al. (2017)*, but the interrelationships of the Delphinida (Lipotidae + Inioidea + Kentriodontidae, as redefined herein + Delphinoidea) are different in the two studies (Fig. 14). As regards Delphinida, the interrelationships of the Delphinoidea (Delphinidae + Monodontidae + Phocoenidae), including the extinct taxa, also recall those recovered

by *Tanaka et al. (2017)*, but the sister group relationships among Delphinida (Lipotidae + Inioidea + Kentriodontidae) are different. *Tanaka et al. (2017)* included in their analysis only three kentriodontids and suggested that they formed a paraphyletic group. These three kentriodontids were located basal to the 'crown' Delphinoidea.

As mentioned above, we performed our phylogenetic analysis by applying a tree constraint based on molecular evidence by *McGowen, Spaulding & Gatesy (2009)*, *McGowen et al. (2011)*; *McGowen et al. (2020)* (Fig. 14) as was also the case for the analysis by *Tanaka et al. (2017)*. *Lambert et al. (2017)* performed their phylogenetic analyses both with a tree constraint based on molecular evidence and without such a tree constraint, and they preferred their unconstrained tree as a result from their multiple analyses. The study by *Lambert et al. (2017)* was so comprehensive that it might be the reason why the molecular evidence had not been used in later studies on the phylogeny of the Delphinida including the kentriodontids (e.g., *Peredo, Uhen & Nelson, 2018*; *Kimura & Hasegawa, 2019*; *Kimura & Hasegawa, 2020*). Molecular phylogenetics is now widely accepted for reconstructing the phylogenetic relationships of organisms, but its results are sometimes different from analyses based only on morphological data. Although many of the aforementioned studies (e.g., *Lambert et al., 2017*; *Peredo, Uhen & Nelson, 2018*; *Kimura & Hasegawa, 2019*) chose the total evidence approach (parsimony analysis based both on molecular and morphological evidence) for their analyses, the resulting relationships they suggested are different from that of the analyses based on molecular data only in regard to the extant species (*McGowen, Spaulding & Gatesy, 2009*; *McGowen et al., 2011*; *McGowen et al., 2020*; *Geisler et al., 2011*; *Post, Louwye & Lambert, 2017*; *Lambert et al., 2018a*; *Lambert et al., 2020*).

Because of relatively low bootstrap values for the result of our phylogenetic analysis, the morphological evidence for the monophyly of kentriodontids is still not robust in support. However, it should be emphasized that the result of our parsimony analysis with tree constraint by molecular evidence is consequently no contradiction with molecular phylogenetics for the extant species.

## Diversifications of Delphinida based on the ear bones

At the time of the evolution and diversification of Delphinida, including Lipotidae, Inioidea, monophyletic kentriodontids, and Delphinoidea, seven out of 18 synapomorphies are considered as evolutionary changes of periotic and tympanic bulla features. These changes could be interpreted as the result of evolutionary innovation, for example the potential specialization of their echolocation abilities among odontocetes (*Gutstein et al., 2014*; *Churchill et al., 2016*). These characters are the following: the processus muscularis of the malleus is sub-equal or longer than the manubrium (Chr. 237), the articulation of the anterior process of the periotic with the squamosal is absent (Chr. 253), the anterior bullar facet is absent (Chr. 254), the dorsal surface of the periotic is nearly flat (Chr. 260), the foramen singulare forms a shared recess with the spiral cribriform tract, the transverse crest that separates it from the proximal opening of the facial nerve canal is low, and the proximal opening of the facial nerve canal is within the internal acoustic meatus (Chr. 269), the aperture for the cochlear aqueduct is smaller than the aperture for the vestibular

aqueduct (Chr. 272), and the dorsal margin of the involucrum of the tympanic bulla is excavated just anterior to the posterior process (Chr. 317). Furthermore, the node uniting the kentriodontids and the delphinoids is supported by 11 synapomorphies, six of which regarding auditory specializations, namely: the apex of the anterior process of the periotic is thickened by the prominent dorsal tubercle that gives to this apex a rectangular section on the plane of the body of the periotic (Chr. 239), the contact of the anterior process of the petrosal with a portion of the ectotympanic bulla anterior to the accessory ossicle is absent (Chr. 249), the periotic articulates with the squamosal along the hiatus epitympanicus and adjacent regions on the posterior process (Chr. 286), the posterior process of the periotic is long (Chr. 292), the lateral furrow of the tympanic bulla is present as a shallow groove (Chr. 303), and the ventral margin of the tympanic bulla is concave in lateral view (Chr. 307). Compared with other odontocetes, the ear bones of delphinidans are highly specialized (e.g., *Fraser & Purves, 1960*; *Gutstein et al., 2014*), and kentriodontids share a number of tympanoperiotic apomorphies with the delphinoids rather than with inioids (see also *Gutstein et al., 2014*). These morphological changes of the periotic and tympanic bulla in the Delphinida are thought to have been emphasized by their diversification or specialization of functional relationships between the periotic, tympanic bulla, and nearby portion of the skull during the process of the acquisition of much higher frequency (i.e., ultrasonic) sound hearing abilities (e.g., *Gutstein et al., 2014*; *Ary, 2017*), and sound reception mechanism (*Cranford, Krysl & Amundin, 2010*). These changes might also have allowed delphinidans to diversify their abilities of echolocation, such as narrow-band and bimodal sound structure (e.g., *Churchill et al., 2016*; *Mourlam & Orliac, 2017*) and habitat preferences (*Costeur et al., 2018*). However, the direct relationship of the above structural changes of tympanoperiotics and resulting functional innovations are still uncertain (*Gutstein et al., 2014*). Therefore, relationships between these morphological and functional changes should be tested through further work. Nevertheless, 13 tympanoperiotic characters out of 29 characters as synapomorphies for the Delphinida still indicate their specialization and innovation of hearing abilities.

Among delphinidans, kentriodontids exhibited a high diversity during the Miocene (*Ichishima et al., 1995*; *Marx, Lambert & Uhen, 2016*, Plate 16a). Based on published records (*Ichishima et al., 1995*; *Ichishima, 1995*; *Dawson, 1996a*; *Dawson, 1996b*; *Bianucci, 2001*; *Kazár, 2005*; *Kazár & Grigorescu, 2005*; *Lambert, Estevens & Smith, 2005*; *Kazár, 2006*; *Whitmore & Kaltenbach, 2008*; *Kazár & Hampe, 2014*; *Salinas-Márquez et al., 2014*; *Peredo, Uhen & Nelson, 2018*; *Kimura & Hasegawa, 2019*); all 31 taxa that can be recognized as kentriodontids are known in the Miocene; six in the early Miocene, 19 in the middle Miocene and six in the late Miocene. Conversely, delphinoids and inioids do not appear until the end of the middle Miocene (*Murakami et al., 2014*; *Murakami, 2016*; *Kimura & Hasegawa, 2020*). In this regard, kentriodontids geochronologically form a first diverse group within the delphinidans, unique for the modifications of their ear bones within the odontocetes. Considering the high ratio of morphological changes observed in their tympanoperiotics and their high species richness, the specializations of their hearing apparatus in kentriodontids probably resulted in their great diversification during the period between the early and middle Miocene.

### Institutional abbreviations

**NMHF**     Ninohe Museum of History and Folklore, Ninohe City, Iwate, Japan.

**NMNS-PV**    Fossil vertebrate collections at the National Museum of Nature and Science, Tsukuba, Japan.

**NSMT-M**    Marine mammal collections at the National Museum of Nature and Science, Tsukuba, Japan.

## ACKNOWLEDGEMENTS

We wish to thank K Sugawara (then NMHF), Y Seki (NMHF) and A Inaba (NMHF) for permitting us to describe NMHF 999. We also thank Y Tajima (NMNS), T Kimura (GMNH) for providing access to the collections under their care. Our thanks also go to Y Tajima and TK Yamada (NMNS), M Oishi (then Iwate Prefectural Museum), H Ichishima (Fukui Prefectural Dinosaur Museum) and Y Yanagisawa (Geological Survey of Japan) for providing useful discussion and advice. We thank K Sashida (then University of Tsukuba, now Mahidol Univ.) and S Agematsu (Univ. Tsukuba) for their help during field works. We are grateful to K Sashida, S Agematsu, K Tanaka (Univ. Tsukuba), and Y Shigeta (NMNS/Univ. Tsukuba) for their useful advice, discussions and generous encouragement during the course of this study. The manuscript has been greatly improved by careful attention to detail and extensive comments from PeerJ editor B Hedrick, reviewers N Pyenson, O Lambert and an anonymous reviewer. We appreciate their efforts very much.

### Funding

The authors received no funding for this work.

### Competing Interests

The authors declare there are no competing interests.

### Author Contributions

- Zixuan Guo performed the experiments, analyzed the data, prepared figures and/or tables, authored or reviewed drafts of the paper, and approved the final draft.
- Naoki Kohno conceived and designed the experiments, performed the experiments, authored or reviewed drafts of the paper, and approved the final draft.

### Data Availability

Raw data is available in the Supplemental Files.

### New Species Registration

The following information was supplied regarding the registration of a newly described species:

Publication LSID: urn:lsid:zoobank.org:pub:B0E9467F-CDD3-4AF4-83FE-40CE09D 15700

*Kentriodon sugawarai* sp. nov. LSID: urn:lsid:zoobank.org:act:0D209916-B472-44A7-B7AB-29682FA945C4.

## Supplemental Information

Supplemental information for this article can be found online at http://dx.doi.org/10.7717/peerj.10945#supplemental-information.

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
