# Peer review of "A new kentriodontid (Cetacea: Odontoceti) from the early to middle Miocene of the western North Pacific and a revision of kentriodontid phylogeny"

_PeerJ, doi:10.7717/peerj.10945_

## Round 0.1 · original submission · Minor Revisions

Thank you for your manuscript. Based on comments from three reviewers, I have found that your manuscript will be publishable in PeerJ following some moderate revisions.

Reviewer #2 suggested a figure showing a summary of different phylogenetic hypotheses. I think that this would really enhance your paper. Your figures are very informative now, but this would likely make your paper more citable.

Also following reviewer 2's suggestions, please expand your discussion of species richness.

Reviewer #3 noted some mismatches between your description, diagnoses, analyses and figures as well. Please go through their comments carefully and check for any errors or inconsistencies that may have impacted your analyses.

Finally, all reviewers suggested changes to the English throughout the manuscript. Please make the changes and if possible, have a native English speaker read the manuscript prior to resubmission.

When you submit your revision, please include a tracked changes copy of the manuscript, a clean version of the manuscript, and a response to reviewers. Please let me know if I can be of any help in the process.

Best,

Brandon

Reviewer 1 ·

Basic reporting

I liked reviewing this paper, which I hope to see published on PeerJ soon. The new taxon description is detailed and careful, and all the figures are excellent. The issues dealt with in Gou & Kohno's manuscript are overall relevant to the palaeocetological community.
Before publication, however, a work of check and revision of the English text is in my opinion strongly needed. English is not my mother tongue, so I can help just a little on this issue; nonetheless, in the attached edited file, I suggested many proposals of edits that mostly (but not exclusively) encompass the English text. The authors might consider having their manuscript language-checked by a native English speaker.
On a more 'personal' note, I feel a bit weird by observing that the diagnosis of Kentriodon has reduced to a single character regarding the size of the external acoustic meatus. Might it be to the rather incomplete nature of K. schneideri, which in your analysis results as the basalmost member of Kentriodon?
Furthermore, some comparisons with Kentriodon hoepfneri might be useful, at least in the diagnosis section.

Experimental design

This is a rigorous and careful description of a new species equipped with a comprehensive phylogenetic analysis that offers new insights on the evolution of kentriodontids.

Validity of the findings

I would only suggest the authors to reconsider the second paragraph of their Discussion: in my opinion, it might be more focused and to the point. The first paragraph of the Discussion is mostly goo but it also might be made more tight and locally refocused (see my suggestions in the attached edited file).

Additional comments

Dear authors,
I must acknowledge the great work you did - I really liked reading your work. I think that your paper can undoubtedly be accepted following moderate revisions. Here I say 'moderate' instead of 'minor' only because the number of corrections I would like to ask you is not negligible, and because a work of check and revision of the English text is in my opinion strongly needed. No doubt you will be able to do that quite easily and in suitable time!
Bests,
the reviewer

Annotated reviews are not available for download in order to protect the identity of reviewers who chose to remain anonymous.

·

Basic reporting

Guo and Kohno report the description of a new species of kentriodontid -- a small, extinct cetacean -- from the middle Miocene of Japan. In general, this manuscript is clear and well written. There are widespread and consistent grammatical mistakes that I've identified in the General Comments (below), which should be fixed. This manuscript provides a straight forward morphological description of an incomplete skull (along with other elements), which serve to underpin the establishment of a novel taxon, specifically a new species in the genus Kentriodon. The osteology is completely described, well illustrated and labeled, and its surrounding context (e.g., stratigraphic context) is well stated. There are no concerns insofar as basic reporting.

Experimental design

The primary analysis was phylogenetic, and the authors do a good job of showing their work -- it builds on recent morphological analyses, and they spell out the source matrices, the relatives strengths, and the specifics of how they improved on past work. In this way, the manuscript provides a solid advancement in understanding the systematics of this long-known (but chronologically understudied) taxonomic group. The analyses and explanations are all straightforward and consistent with general reporting for these details among practicing fossil cetacean systematicians.

Validity of the findings

The manuscript is weakest on the underlying arguments about the evolution of Kentriodontidae. The authors should first state clearly what the general historical thinking has been, and be clear about the studies: some considered them paraphyletic stem delphinoids (in the modern parlance, though not at the time of the studies); others did indeed propose a monophyletic Kentriodontidae sister to Delphinoidea. Others still argued for kentriodontids being polyphyletic. A real contribution in this regard would be a nice figure showing a summary of the different phylogenetic hypotheses, even in a simplified format. Geisler et al. 2011 does this well, for example. After all, the novel contribution of this study is the support for a monophyletic Kentriodontidae and genus Kentriodon.

The authors are also weak on matters of species richness. How many kentriodontids through time are there? Simply saying "a rich diversity" is vague and not compelling. It is easy to generate such a plot at any taxonomic level using data from the Paleobiology Database -- which can be cited as such. Are they more or less rich, per unit of geologic time, than other odontocetes? More or less than crown delphinoids? The authors owe the readers a bit more on these answers to explain why kentriodontids are interesting, but have largely escaped detailed study.

Lastly, the connection between tympanoperiotic specializations for hearing and diversification are unfounded, unclear and speculative. Speculation is fine, but there's not logic nor specifics to WHY some kinds of suites of tympanoperiotic specializations would confer a given functional result. Some have recently tried to do this with external morphology (e.g., Gutstein et al. 2014), or with internal microCT scans of the internal laminae of the cochlea. But not one has drawn a connection between anatomy and diversification -- why? Why not? The authors provide no details, nor any about relative orgination and extinction rates among Delphinida that would support these claims. Overall, this speculation comes off unpersuasive. What would be a testable hypothesis here? The authors should ponder that.

Additional comments

Specific comments:

Throughout the manuscript, the full taxonomic name and authority citation needs to be only stated once, at first mention -- subsequent mentions do need need to use the full author and year.

l. 21, Change to "consists" from "is consisted"

l. 23, Add "a" to "shares a unique"

l. 24, What does "it" refer to? The character state or the taxon? Please clarify.

ll. 25-26, Add "a" to "a dorsolateral edge" and "that" to "foramen that is formed" and edit the last item as "and three or more anterior dorsal infraorbital foramina."

ll. 28, It's well said in the Discussion section; suggest editing as "yielded a consensus showing all previously identified kentriodontids as a monophyletic..."

Ll. 30-31, What's a dynamic renewal? I think it's just a character state reversal. Please clarify and do not embellish.

l. 36, "most diversified" is a bit clunky -- "most species-rich group of living marine mammals" might be more appropriate. "Origin and adaptation" is also an antiquated way of talking about studying evolutionary origins. I would say "evolutionary origins are still a puzzle" instead. A puzzle is a metaphor, not a literal construct.

l. 39, There are better references (and more recent ones) for the high diversity of delphinioids in the late Neogene -- please cite 21st century references and be specific "high species richness." Taxonomic richness is just one measure of diversity.

l. 41, Edit: characteristics or character states, but not characters.

l. 42, Edits: nasals (plural); and premaxillae -- not premaxillary bones

l. 43, "retained in kentriodontids" implies a polarity of character states that should be more clear -- many living phocoenids have symmetrical facial bones.

l. 44, Edit: evolutionary patterns (plural) -- because there's likely more than just one! Morphology, body size, origin and extinction, etc.

ll. 47, 49, 64, 66, 112, 503, 571, 574, delete "the" from "the kentriodontids"

l. 50, Add "a" to "a different combination"

l. 51, Add "in to "in a broad sense"

l. 59, Please choose another verb than "rectify" -- I am not sure what's meant here. The authors of that study simply do not recover a monophyletic Kentriodontidae. The grammar needs to be fixed throughout this sentence.

l. 62, Edit: "in a different phylogenetic typology." These words will help with the precision of the sentence.

l. 63, Edit: "the relationships among Delphinoidea are still"

l. 66, It's unclear how you reassess the relationships with a molecular consensus -- can you be more precise. The verb-subject agreements and grammar of this sentence need to be fixed.

ll. 92, 96, 115, Edit: "X kentriodontid taxa" instead of "X taxa of kentriodontids"

l. 102, Please do not use "'lower'" in quotes nor in any euphemistic way. If basal branching clades are meant, then say so.

l. 104, Which two streams of studies? It is very unclear, even after re-reading several times. Please state it once again, for the reader's benefit. This sentence also requires some preposition maintenance, especially on l. 107: "far too different from each other."

ll. 117-132, This entire paragraph needs careful scrubbing for the distinction between rock-time and time units: upper Miocene is not the same thing as late Miocene, and the authors need to be more clear about provenance from geography versus stratigraphy. Essentially, all of the stated kentriodontid taxa need be harmonized for these distinctions.

l. 152, Edit heading as "Emended Family Diagnosis" -- but why is the rank of family being used here when the preceding systematics are rank-free clades? I suggest simply saying ""Emended Diagnosis of Kentriodontidae," for simplicity.

l. 153, Edit: "character states"

ll. 162, 469, It is incorrect to use "circa" as an equivalent word for "approximately." Circa is a temporal word to refer to chronological time -- not distance.

l. 170, Does "including" really need to be abbreviated in an online open-access journal? I think not.

l. 196, Edit: "the 1940s" and edit "from a locality close to the Mabechi River"

l. 201, Edit: "a diatom flora" or more clearly "diatoms"

l. 206, Edit: again, "richness of diatoms"

l. 207, The Denticulopsis praelauta Zone number must be stated, if it exists. It was not identified in l. 201.

l. 201, Typo of Desmostylus

l. 212, Edit: "the suggested environment"

l. 218, delete the word "away" it is not necessary. "Choanae" are plural, thus it must be "choanae are cracked"

l. 285, Typo, bony nares.

l. 329, delete the word "away" it is not necessary

l. 443, Question: are there any measurements possible for C1? An OCB measurement would permit a body size estimate.

l. 457, Typo: Delphinidae

l. 458, Misspelling and also: did any one accuse kentriodontids of being ancestral to delphinoids? I suspect not -- no modern worker writes in this way. The issue had always been whether they were a sister group or paraphyletic stemward of crown delphinoids.

ll. 476-483, The genus is monophyletic, according to the analyses. Worth stating!

l. 496, Edit, "so-called"

l. 497, Edit, "their" not "there"

l. 498, Edit, "i.e." is poorly used here -- it means "as in," not "e.g.," which means for example. Here, it would be much more effective to say "such as" or "e.g."

l. 500, Edit: "different from molecular phylogenetic analyses"

l. 509, 529 Which diagram? A figure in this manuscript? Be more precise.

ll. 544-561. Do any of these characters have more or less to do with specialization of hearing, relative to other odontocetes? Otherwise, this list is a perhaps not very compelling. Using the word "peculiarities" on l. 564 is not sufficiently precise and it should be explained and identified better. What features are specialized for kentriodontids and how did that make them different from co-occurring taxa?

l. 572-574, Please edit this sentence for clarity, especially minding prepositions.

l. 574, Why the speculation? What is the basis for supposing hearing specializations "probably resulted" in their high diversification? This is not compelling nor even reasoned as a hypothesis.

Figs. 10-11, Typo in the caption: "holotype," not "holotypes"

Fig 12. Please scrub the figure for typos. For example Physeteridae or Physeteroidea, not the misspelled Physeteridea. Also ZMT 73 should not be italicized. Also, for the caption, why is consensus surrounded by quotes? That is not correct nor warranted -- and the meaning needs to be explained.

Lipotes and Parapontoporia do not fall in a traditional Inioidea concept of Iniidae+Pontoporiidae, so please correct the delimitation of Inioidea in the figure.

Table 1. "premaxillaries" is not an osteological element. Please correct to "premaxillae"

·

Basic reporting

The text is concise and rather well written, but would probably benefit from the review of a native English speaker. I made a series of suggestions to improve the style and clarity, but some more work may be needed. The background is properly presented, and the authors took account of a reasonable amount of previous papers, which are relevant and coherently cited. The work is finely organized, although I made a few suggestions for minor changes. Figures are of very good quality, showing a lot of details. I added some comments for additional labels and for the correction of some terms that are not correctly spelled or do not point to the right feature.

Experimental design

This research is original, dealing with the description of a new species based on a new fossil specimen. It falls definitely in the scope of PeerJ. The main goals of the work and the methods are finely described, and the analyses have been performed rigorously.

Validity of the findings

The description of this very nice new specimen is good, detailed, but includes a few anatomical interpretations that should probably be revised. The systematic and phylogenetic conclusions of the study are well supported. Please find below one concern about the attribution of the new species to the genus Kentriodon and some suggestions to improve the diagnosis and comparison. Some parts of the discussion should be slightly modified to better take account of the relationships between lipotids, inioids, and delphinoids (see annotated pdf).

Additional comments

I made a large number of comments and suggestions in the text and directly on the figures, but please find below several somewhat more important issues:

1. In the title you state 'middle Miocene' for the geological age of the specimen, but in figure 1 its lithostratigraphic interval seems to range between late Burdigalian and early Langhian. With such an interval it would probably be better to write 'late early to earliest middle Miocene'. I also made some comments in this issue in the pdf.

2. The only character defining here the genus Kentriodon (narrow external auditory meatus) is definitely difficult to code based on data from the literature, and has until now, to my knowledge, never been properly quantified. From my personal experience I do not remember having observed significant differences at this level between 'kentriodontids'. I understand the advantages of providing diagnoses based on the phylogenetic analysis (although it makes it difficult to tell if the observations were made by you or if you extracted the data from codings in previous matrices), but in this case, I believe that a differential diagnosis including all other kentriodontids should be provided, to better support the referral of the new species to the genus Kentriodon.

Furthermore, the external auditory meatus does not seem to be preserved in the holotype of the new species (for example I cannot see the postglenoid process of the squamosal), which would make its referral to the genus Kentriodon rather weak. This should be checked, before maybe revising the diagnoses.

I noted several other mismatches between the description, figures, diagnoses, and phylogenetic analysis (for example the dorsolateral margin of the ventral infraorbital foramen), that would necessitate some clarification (see pdf).

3- The only elements of comparison are found in the diagnosis, which is mostly based on the phylogenetic analysis (see comment above). I would suggest either adding comparison elements in the description part or adding a separate comparison section after the description, focusing on similarities and differences with other kentriodontids, especially members of the genus Kentriodon (but not only), at the level of the vertex, periotic, tympanic, malleus... This is a beautifully preserved specimen, providing interesting morphological features that should in my opinion be confronted to other early delphinidans. This would most likely help supporting the diagnosis of the new species and also provide an idea of intrageneric variation.

Even at a broader scale, differing from several previous analyses kentriodontids fall here as later branches compared to lipotids and inioids, and one would have expected elements of comparison with these two clades, which include a continuously improving fossil record (especially inioids). In other words, it may be worth showing how kentriodontids are more derived (towards delphinoids) than inioids. This is only a suggestion, but it would provide further support to your conclusions, I think.

I am looking forwards to seeing this interesting work being published.

O Lambert

---

## Round 0.2 · Minor Revisions

Dear authors,

Thank you for your submission to PeerJ. I appreciate your careful attention to reviewer comments and believe this manuscript will be acceptable pending these few last changes suggested by the reviewer. I have also noted a few grammatical errors that need to be corrected prior to publication (below). Once these have been completed, I will put push the manuscript on to the proof stage. It will not need another round of reviews.

When you submit your revision, please include a clean version, tracked changes version, and reviewer response. Thank you for your submission.

Best,

Brandon Hedrick, Ph.D.




Line numbers correspond with tracked changes version

Line 72: I’m not sure what you mean by ‘as in a broad sense’. Rephrase?

Line 74: ‘have been recently reported’

Line 680: Change to ‘Regarding the kentriodontids,’

Line 1206: ‘ventral’ spelling

Line 1829: ‘beside’

Line 2210: ‘because of the deformation’

Line 2238: Change to: ‘Although it is broken, the postorbital process is’

Line 2245: space between broken and the comma

Line 2555: ‘likely to be’

Line 2561: ‘extends’?

Line 2851: Grammar, rephrase

Line 2876: ‘anteriormost’

Line 2879: ‘cribriform’

Line 3266: ‘different’?

Line 3357: Capitalize ‘Kentriodon’

Line 3427–28: Grammar, rephrase.

Line 3538: ‘However,’ instead of but

Line 3552: ‘cribriform’

Line 3659: ‘did not’ rather than ‘almost never’?

Line 3747: ‘B. Hedrick’ rather than ‘H. Brando’

·

Basic reporting

see below

Experimental design

see below

Validity of the findings

see below

Additional comments

I would like to thank the authors for having made the effort of addressing many comments and suggestions by the reviewers. I think that the way they handled the reviews lead to a significant improvement of their interesting work.
Due to time constraints I wrote all my comments and suggestions in the .doc version of the text AND the .pdf (for figure captions and labels in figures).
- I mostly noted a number of very minor issues with style and a few typos, which should be easily corrected.
- Only a few somewhat more important problems remain, and most deal with anatomical interpretations (concerning the squamosal, exoccipital and basioccipital).
- The new figure 1 is very useful, but some changes are needed in the corresponding caption.
- Concerning the phylogeny, it seems that you missed a couple of recent papers for the comparison of the obtained topology (see suggestions in the text).
I am looking forwards to seeing this fine paper being published.

---

## Round 0.3 · accepted · Accept

Dear authors,

Thank you for your corrections. I enjoyed reading your paper and loved your figures. I am now happy to accept your manuscript for publication in PeerJ. Prior to publication, I did note several small grammar issues that should be corrected (outlined below).

Please let me know if you have any questions.

Best,

Brandon P. Hedrick, Ph.D.



Line 31: I think ‘renewal’ must be the wrong word here, but I am not sure what you are going for. Please reword this sentence.

Line 168: ‘unique combination of characters’

Line 247: ‘weakly’ rather than weekly

Line 343: ‘left squamosal only preserves the postglenoid’

Line 345: ‘squamosal is likely to be long’

Line 517: ‘Since the nasal septum is fragile’